# τBEN: BENCHMARKING AND TESTING THE RULE-BASED TEMPORAL LOGIC REASONING ABILITY OF LARGE LANGUAGE MODELS WITH DATALOGMTL

## ABSTRACT

Large language models (LLMs) are increasingly adopted for a variety of tasks, including multi-hop question answering, knowledge probing, and symbolic commonsense reasoning. While LLMs have advanced the state-of-the-art in these areas, their ability to explicitly solve temporal logic reasoning problems—a complex cognitive process involving the understanding, representation, and manipulation of temporal information such as events, their durations, and relationships—remains largely unexplored. To enhance understanding of LLM performance in this common task widely explored in the traditional symbolic AI field, we have developed a new set of synthetic benchmark for rule-based temporal logic reasoning named τBEN. τBEN is developed within the context of DatalogMTL, a powerful knowledge representation language designed for reasoning about the properties of systems that evolve over time. Notably, this benchmark defined six levels of rule complexity and provides flexible configurations, allowing for the customization of temporal rules and task complexity to suit various needs.

We evaluated the close-sourced GPT-4o and the open-sourced Llama-3[1] using three common prompting settings—*zero-shot*, *few-shot*, and *zero-shot-CoT*—on our synthetic benchmark. Our key findings are as follows: (i) Without generating the reasoning process (chain-of-thought), even advanced LLMs like GPT-4o exhibited nearly random performance on these rule-based temporal logic reasoning tasks. However, with chain-of-thought prompting, LLMs demonstrated preliminary temporal logical reasoning abilities; (ii) Both GPT-4o and Llama-3 were unable to solve temporal logical reasoning problems involving recursion, indicating a lack of advanced complex reasoning capabilities in understanding symbolic representations involving time; (iii) There is significant room for improvement in leveraging large language models to address problems widely explored in the traditional logic-based AI domain. Prompts and datasets are available in the appendix, and a datasheet for τBEN is also provided.

## 1 INTRODUCTION

Time is a crucial and pervasive concept in our daily life and the real world that requires management in various ways: events happen at specific moments, certain facts are valid over a time period, and temporal relationships exist between facts and events Ozsoyoglu & Snodgrass (1995). Time must be considered when representing information within computer-based systems, querying information about the temporal features of the real world, and reasoning about time-oriented data. In traditional logic-based AI (also known as symbolic AI), researchers have developed a variety of rich knowledge representation rule languages, such as Linear Temporal Logic(LTL) Huth & Ryan (2004), Signal Temporal Logic (STL) Donzé (2013) and DatalogMTL Brandt et al. (2018) to represent and reason these timestamped events. he syntax, semantics, and complexity of these rule languages, along with their corresponding solvers, have been well developed over the past several decades.

In recent years, there is evidence that current large language models (LLMs) can perform fundamental data analysis tasks Reid et al. (2024); Bai et al. (2023) such as visualization Maddigan & Susnjak

---

[1]We experimented with Meta-Llama-3-8B-Instruct.

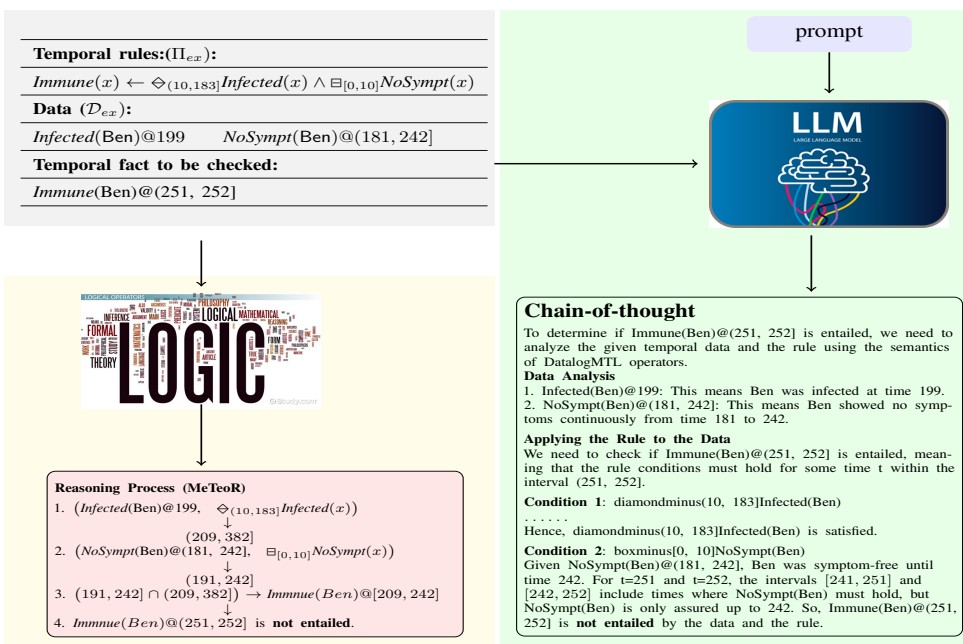

Figure 1: Overview of two approaches (symbolic reasoner based and LLMs based) for solving the temporal reasoning. The former can use an existing symbolic reasoner (e.g., MeTeoR Wang et al. (2022)) to derive temporal facts, which are then used to judge whether the given facts are entailed. In contrast, the latter leverages the power of large language models to complete the reasoning process and provide the answer. In particular, we need to provide some background knowledge about the semantics of used notations, which is included in the prompt.

(2023); Cheng et al. (2023), commonsense reasoning Wang & Zhao (2023a); Zhao et al. (2024) and mathematical reasoning Ahn et al. (2024). However, little effort has been made to evaluate the more challenging aspects of rule-based temporal logic reasoning, a long-standing research problem in traditional logical artificial intelligence. While some studies have benchmarked or evaluated the temporal reasoning abilities of LLMs Wang & Zhao (2023b); Xiong et al. (2024), they primarily focus on reasoning over temporal data expressed in natural language, without addressing the temporal logic, which is typically represented as logical rules with well-established syntax and semantics.

In this paper, we evaluate the rule-based temporal logic reasoning abilities of large language models (LLMs) from a logical perspective, considering the semantics of temporal logic. Our aim is to bridge the gap between traditional logic-based AI and LLMs. Specifically, we focus on the classic temporal reasoning problem known as *fact entailment* Cheng (1996); Brandt et al. (2018); Wałęga et al. (2023b). In our evaluation task, we consider temporal data of the symbolic form $P(a_1, \ldots, a_n)@\varrho$, where $P$ denotes a predicate (relation), $a_i$ is an entity, $n$ denotes the arity[2] and $\varrho$ represents a punctual time point or time interval. Given a set of temporal rules and a target temporal fact, the task is to determine whether the fact is entailed by the temporal data and rules. To provide better intuition, we use Example 1 togeter with Figure 2 to describe the problem.

**Example 1.** *There is growing evidence that individuals develop COVID-19 immunity if they were infected within the last 6 months (discounting the last ten days when they had no symptom) Feikin et al. (2022). The condition can be captured by a DatalogMTL program $\Pi_{ex}$ with the following rule:*

$$Immune(x) \leftarrow \diamondsuit_{(10,183)}Infected(x) \;\wedge\; \boxminus_{[0,10]}NoSympt(x)$$

---

[2]If the arity is 0, then $P$ is treated as a statement that is either true or false. This differs from temporal knowledge graphs, which consist solely of quadruples (arity=2).

*The above rule checks whether an individual infected at some point in the last six months excluding the last 10 days (operator $\diamondplus_{(10,183]}$) remained continuously without symptoms in the last 10 days (using the 'box past' operator $\boxminus_{[0,10]}$).*

*Then, we assume a dataset contains some historical data about a person called Ben in the form of facts stamped with validity intervals, where the first day of the year is given by the interval $(0, 1]$, the second day by $(1, 2]$, and so on. Ben got vaccinated at July 19 (represented as $199$). Moreover, Ben had no symptoms since July 1 (i.e., $181$) until August 30 (i.e., $242$). This is represented by a dataset $\mathcal{D}_{ex}$ with the following facts:*

$$Infected(\mathsf{Ben})@199, \quad NoSympt(\mathsf{Ben})@(181, 242]$$

*If we want to know whether Ben is immune between September 8 and September 9, represented as a temporal fact $Ben@(251, 252]$, we can formulate this as a* fact entailment *problem: Is $Ben@(251, 252]$ entailed by $\mathcal{D}_{ex}$ and $\Pi_{ex}$?*

Traditionally, a symbolic reasoner is used to check entailment by applying temporal rules to temporal data, deriving new facts, and verifying if the given fact is among the derived ones. As shown in Figure 1, this process involves several steps before reaching a conclusion. We are exploring whether large language models (LLMs) can solve such problems by providing correct answers along with the reasoning process and human-readable textual explanations. On one hand, evaluating the temporal reasoning capabilities of LLMs could complement evaluations of other reasoning skills, such as mathematical and commonsense reasoning. On the other hand, using LLMs as a tool or a complementary tool for temporal reasoning alongside symbolic reasoners could significantly contribute to the development of traditional logic-based AI.

Our contributions are summarized as follows:

- We propose a new benchmark, TBEN, to test the rule-based temporal logic reasoning abilities of large language models through synthetic tasks. Our benchmark generator allows for the creation of benchmarks with varying data sizes and rule structures of differing complexity, enabled by flexible configurations.

- We conduct preliminary experiments using two common large language models on our constructed benchmarks. Our results demonstrate that, without advanced prompting techniques such as Chain-of-Thought (CoT), even GPT-4o performs almost at random. However, with CoT, GPT-4o shows a significant performance boost, while open-source models like Llama-3-8b continue to exhibit comparatively lower effectiveness in solving rule-based temporal logical reasoning problems.

- We further conducted extensive analysis, including a detailed ablation study, natural language-based[3] temporal logic reasoning experiments, error analysis, and human evaluation. Our analysis and key findings can serve as a foundation for LLM rule-based temporal logic reasoning benchmarking, and motivate further work to enhance these capabilities.

We will open source our developed TBEN to spur future research in improving the temporal logic reasoning ability of large language models.

## 2 DATALOGMTL

DatalogMTL Brandt et al. (2018); Wałęga et al. (2019) is a powerful knowledge representation language, which extends Datalog Abiteboul et al. (1995), a widely used declarative logic programming language, with operators from metric temporal logic (MTL) Koymans (1990). Different Datalog designed to handle static facts and rules due to lack of built-in temporal constructs, DatalogMTL equipped with MTL operators is enabled to reasoning about properties of systems that evolve over time. These operators build upon the standard linear temporal logic (LTL) Huth & Ryan (2004) operators, such as $\diamondminus$ standing for "sometime in the past", $\boxminus$ for "always in the past", and $\mathcal{S}$ for "since",

---

[3]Manually converting each rule into natural language requires substantial human effort. Therefore, we propose an alternative approach: using customized templates to verbalize the temporal rules. The verbalized examples could be found in the appendix.

as well as their future counterparts $\oplus$ for "sometime in the future", $\boxplus$ for "always in the future", and $\mathcal{U}$ for "until". In MTL, however, these LTL operators are annotated with intervals; for instance, the expression $\ominus_{[1,2]} LiveIn(x, y)$ is true at time $t$ if entity $x$ lived in location $y$ sometime between times $t - 1$ and $t - 2$. Similarly, $\boxminus_{[1,2]} LiveIn(x, y)$ holds at time $t$ if $x$ continuously lived in $y$ throughout the aforementioned time interval. In this section, we recapitulate the syntax, semantics, and key temporal tereasoning problems in DatalogMTL.

**Syntax**  We consider a *signature* consisting of pairwise disjoint countable sets of constants, variables, and predicates with non-negative integer arities. A term is either a constant or a variable. A *relational atom* is an expression of the form $P(\mathbf{s})$, with $P$ a predicate and $\mathbf{s}$ a tuple of terms whose length matches the arity of $P$. In this paper, we restrict ourselves to a fragment in which metric atoms are generated by the following grammar, where $P(\mathbf{s})$ is a relational atom and $\varrho$ an interval including only non-negative numbers:

$$M ::= P(\mathbf{s}) \mid \ominus_{\varrho} M \mid \oplus_{\varrho} M \mid \boxminus_{\varrho} M \mid \boxplus_{\varrho} M$$

A rule in this fragment is an expression of the form

$$P(\mathbf{s}) \leftarrow M_1 \wedge \cdots \wedge M_n, \qquad \text{for } n \geq 1, \tag{1}$$

where the body atoms $M_1, \ldots, M_n$ are metric atoms and the head atom $P(\mathbf{s})$ is relational. A program is a finite set of rules.

**Semantics**  An interpretation $\mathfrak{I}$ is a function assigning truth values to ground relational atoms $P(\mathbf{c})$ and time points $t \in \mathbb{Z}$. It determines if $P(\mathbf{c})$ is satisfied at $t$, denoted as $\mathfrak{I}, t \models P(\mathbf{c})$, or not, denoted as $\mathfrak{I}, t \not\models P(\mathbf{c})$. This notion of truth assignment extends to other ground metric atoms in the considered fragment as follows:

$$
\begin{aligned}
\mathfrak{I}, t &\models \ominus_{\varrho} M &\quad \text{iff} \quad& \mathfrak{I}, t' \models M \text{ for some } t' \text{ with } t - t' \in \varrho, \\
\mathfrak{I}, t &\models \oplus_{\varrho} M &\quad \text{iff} \quad& \mathfrak{I}, t' \models M \text{ for some } t' \text{ with } t' - t \in \varrho, \\
\mathfrak{I}, t &\models \boxminus_{\varrho} M &\quad \text{iff} \quad& \mathfrak{I}, t' \models M \text{ for all } t' \text{ with } t - t' \in \varrho, \\
\mathfrak{I}, t &\models \boxplus_{\varrho} M &\quad \text{iff} \quad& \mathfrak{I}, t' \models M \text{ for all } t' \text{ with } t' - t \in \varrho.
\end{aligned}
$$

For example, an interpretation making atom $LiveIn(Ann, Paris)$ true everywhere within $[10, 30]$ and false elsewhere makes $\boxminus_{[1,2]} LiveIn(Ann, Paris)$ true at the time point 31, but false at 32. An interpretation can be alternatively seen as the (possibly infinite) set of facts that it satisfies, which yields a natural meaning to containment and minimality of interpretations.

## 2.1 Major Temporal Reasoning Problems

According to Brandt et al. (2018); Wałęga et al. (2019), temporal logic reasoning involves two major problems: *consistency checking* and *fact entailment*. *Consistency checking* is the task of determining whether a given program and dataset admit a common model Emerson (1990); Schnoebelen (2002). *Fact entailment* involves checking whether a program and dataset together entail a specific relational fact. Brandt et al. (2018) note that in $DatalogMTL$, consistency checking and fact entailment are complementary problems. Consequently, this paper focuses solely on the *fact entailment* problem to evaluate the temporal reasoning capabilities of large language models.

## 3 tBen: A Rule-based Temporal Logic Reasoning Benchmark with DatalogMTL

DatalogMTL is a powerful knowledge representation language that can characterize complex temporal conditions by defining various rules using combinations of different atoms and temporal operators ($\ominus, \oplus, \boxminus, \boxplus$) whose semantics has been described in Section 2. To some extent, the complexity of a *fact entailment* problem is largely determined by the complexity of associated temporal rules.

**Challenges**  Although DatalogMTL is a powerful knowledge representation language, editing such rules requires considerable expertise and is very time-consuming. Currently, there are no suitable benchmarks in terms of size and rule diversity available for research. In the era of large language

models (LLMs), we are motivated to explore whether LLMs can assist in addressing temporal reasoning problems traditionally solved by symbolic reasoners. Therefore, developing data and rule generators that enable the automatic construction of benchmarks with flexible configurations for customized rule structures and task complexity is important and poses a substantial challenge.

---

**Zero-shot Prompt Prefix**

Given a dataset, temporal rules and a temporal fact, you need to apply the rules to the dataset and then judge whether the given fact is entailed by the dataset and rules.

The rules are expressed as DatalogMTL, a language of temporal logic that extends Datalog with operators from metric temporal logic (MTL). The semantics of four MTL operators are given as follows:

If $\diamondsuit_{[a,b]}A$ is true at the time t, it requires that A needs to be true at some time between t-b and t-a.

If $\boxminus_{[a,b]}A$ is true at the time t, it requires that A needs to be true continuously between t-b and t-a.

If $\diamondsuit[a,b]A$ is true at the time t, it requires that A needs to be true at some point between t+a and t+b.

If $\boxplus[a,b]A$ is true at the time t, it requires that A needs to be true continuously between t+a and t+b.

Now, we have a data, some DatalogMTL rules and a fact entailment question. You should only output true or false, and please do not output other words.

---

**SingleAtom**

**R**: $A \leftarrow \diamondsuit_{[1,2]}B$
**D**: $\{B@[4,5]\}$
**Q**: A@6 is entailed?

**MultiAtoms**

**R**: $A \leftarrow \boxminus_{[3]}B \wedge \boxminus_{[2,\ 3]}C$
**D**: $\{B@[1], C@[1,\ 3]\}$
**Q**: A@3 is entailed?

**Rational**

**R**: $A \leftarrow \boxminus_{[1.2,\ 2.1]}B \wedge \boxminus_{[4.2,\ 5.1]}B$
**D**: $\{A@[1.1]\}$
**Q**: A@[2.4,\ 2.8]) is entailed?

**MixedOperators**

**R**: $A \leftarrow \diamondsuit_{[1,\ 2.4]}B \wedge \boxplus_{[1,\ 2]}C$
**D**: $\{B@[1], C@[2,\ 4]\}$
**Q**: A@2.3 is entailed?

**MultiRules**

**R**: $D \leftarrow \boxminus_{[2]}B \wedge \boxplus_{[1,2]}C$
$A \leftarrow \diamondsuit_{[1.5,\ 2]}D \wedge \diamondsuit_{[2]}C$
**D**: $\{B@[1], C@[2,5]\}$
**Q**: A@[4.5,5] is entailed ?

**Recursive**

**R**: $A \leftarrow \diamondsuit_{[1,2]}A \wedge \boxminus_{[1,10]}C$
**D**: $\{A@[1], C@[1,100]\}$
**Q**: A@99 is entailed?

Figure 2: Overview of the TBEN Benchmark, featuring six levels of temporal reasoning problems with varying complexity. We present an intuitive example representing each level, along with the corresponding rule, dataset, and fact entailment problem. A zero-shot-prompt prefix is also provided (see Appendix for additional prompt prefixes used in this paper). For better demonstration, we use the symbols $\diamondsuit$, $\diamondsuit$, $\boxminus$, and $\boxplus$, which are replaced by $<->$, $<+>$, $[-]$, and $[+]$, respectively, in the actual prompts due to typing constraints.

## 3.1 Leveling DatalogMTL Rules

To address the aforementioned challenge and provide a more comprehensive evaluation of the temporal reasoning abilities of large language models, we aim to create a new synthetic benchmark with flexible configurations for customizing rule structures and task complexity. We classify DatalogMTL rules into six classes (**SingleAtom**, . . . , **Recursive**) based on their structural representations, considering factors such as the number of body atoms, the number of temporal operators used, the number of rules involved, and whether the rules are recursive. While we are unable to quantify the degree of complexity of each level, we assume that higher levels correspond to greater complexity. This assumption is based on the observation that more complex rule structures require additional temporal reasoning steps when using a symbolic reasoner like MeTeoR Wang et al. (2022).

**SingleAtom**  The most simplest form of a rule is $A \leftarrow \oslash_{[\rho]}B$, where $\oslash$ could be one of the four metric temporal operators ( $\boxminus$, $\boxplus$, $\diamondsuit$ and $\diamondsuit$). In particular, we ensure that $A$ and $B$ are two different atoms, so only one calculation operation. A **SingleAtom** example is given in Figure 2, where we can derive $A@[5,7]$ based on the given dataset and the rule, entailing that $A@6$ is true. In particular, we consider DatalogMTL over the integer timeline, a fragment of DatalogMTL Wałęga et al. (2020) and use one type of MTL operator.

**MultiAtoms**  In the **SingleAtom**, the body contains only one atom, so a single rule application is sufficient to complete the derivation. In **MultiAtoms**, we increase the number of atoms in the rule body, requiring not only the validation of each atom but also an intersection operation to obtain the final valid interval. As the example shown in Figure 2, the rule contains two atoms. First, we calculate the valid intervals for each atom. Based on the provided temporal facts, $\boxminus_{[3]}B$ holds only

at the punctual time point $[4, 4]]$, and $\boxminus_{[2,3]}A$ holds at the interval [4,5]. The intersection of these intervals, $[4, 4]$ and $[4, 5]$, is $[4, 4]$. Thus, we derive that $A$ is true at the time point 4, so $A@4$ is entailed. As with **SingleAtom**, we consider DatalogMTL over the integer timeline Wałęga et al. (2020) and use only one type of MTL operator.

**Rational**   Both **SingleAtom** and **MultiAtoms** focus solely on the integer timeline, which represents a relatively limited time space and simplifies reasoning due to the integer semantics Wałęga et al. (2020). In **Rational**, we build on top of **MultiAtoms** by expanding the timeline to include the rational numbers, incorporating decimal time points. Intuitively, rational-based numerical operations are more complex than their integer-based counterparts, and we aim to determine if large language models exhibit similar behavior. We continue to use only one type of MTL operator at this level.e type of MTL operator in the level.

**MixedOperators**   Using only one operator limits the expressiveness of DatalogMTL, preventing the definition of complex temporal conditions. Thus, a natural expansion is to allow the use of MTL operators. The four types of MTL operators can be used to define temporal conditions associated with both the past and the future. A **MixedOperators** example is shown in Figure 2, which involves two MTL operators ($\diamondsuit$ and $\boxplus$). To complete the derivation, we first calculate the valid interval where $\diamondsuit_{[1,2.4]}B$ with the past operator ($\diamondsuit$) holds, which is $[2, 3.4]$. Then, we calculate $\boxplus_{[1,2]}C$, whose valid interval is $[1, 2]$. After performing the interval intersection, we obtain that A holds at the time interval $[2, 2]$. Thus, the temporal $A@2.3$ is not entailed.

**MultiRules**   In the previous four levels, fact entailment is associated with only one temporal rule. However, in more practical scenarios, multiple temporal rules may be required to express complex temporal conditions. In this level, we consider a multi-rule temporal reasoning case, where fact entailment involves multiple temporal rules and rule applications must be executed across these rules to complete the derivation. As the example in Figure 2, to derive the target atom $A$, we need to know both $D$ and $C$. However,the dataset only provides the information about $C$. We can derive the $D$ holds at 3 according to the first temporal rule $D \leftarrow \boxminus_{[2]} \wedge \boxplus_{[1,2]}C$; then, we can derive that $A$ holds at the interval $[4.5, 5]$ according to the second rule. Hence, $A@[4.5, 5]$ is entailed.

**Recursive**   The fact entailment problem at this level is considered the hardest because it involves recursion. Unlike static knowledge representation languages (e.g., Datalog), where all facts can be derived after a certain number of rule applications, some recursive rules in DatalogMTL may require an infinite number of applications. Even for symbolic-based approaches, this presents a significant challenge, and researchers have devoted considerable effort to addressing it Wałęga et al. (2021; 2023a). According to Wałęga et al. (2023a), in the recursive scenarios, periodic structures will ultimately occur repeatedly, but calculating these periodic structures is challenging. From a human perspective, however, identifying such periodic structures can be straightforward. For instance, consider a recursive rule $\boxplus_{1year}\text{Bday}(x) \leftarrow \text{Bday}(x)$, which states that anyone having their birthday at a time point t will also be having their birthday at the same time the following year. If we know that Ben has his birthday on Jun 8, 1991, it is easy to know that he will have his birthday on Jun 8, 1992, Jun 8, 1993 and so on. However, this is difficult for traditional symbolic-based approaches to handle. Therefore, we design fact entailment problems associated with recursive rules to test whether large language models can perform well in this setting.

Specifically, we use facts from both propositional logic Klement (2004) and first-order logic Barwise (1977). The former contains declarative statements that are either 'true' or 'false', while the latter includes expressions with one or more variables. For example, we allow both forms of temporal facts: $Raining$ and $Immune(x)$. The former states that an event (raining) is occurring, while the latter denotes that a property (immune) is associated with an entity, where $x$ acts as a placeholder that can be instantiated to any entity, such as $Immune(Ben)$, indicating that Ben is immune.

### 3.2 GENERATING TEMPORAL DATA AND RULES

The benchmark generation process can be mainly divided into the following three steps: 1) Graph construction, 2) Data generation, and 3) Rule generation. The pseudocode for this benchmark generation algorithm can be found in Part F of the appendix.

**Graph construction**    We employ a general-purpose random graph generator to generate a connected directed random graph. The nodes in the random graph represent predicates, such as A, B, and C. Each edge in this graph represents a body atom of a rule pointing to the corresponding head in the rule. In particular, a predicate can appear in bodies of multiple different rules.

**Data generation**    After the construction of the graph, the program will traverse each nodes in the graph and randomly assign time points or time intervals to the chosen nodes. The time points or intervals are generated based on a given range.

**Rule Generation**    Once the temporal data is generated, the rule generator traverses the edges of the graph, assigning random operators and intervals to the edges. To ensure the generated graph is non-trivial, a reasoning process is performed across the entire graph after completing this step to ensure new facts can be inferred. If multiple rules are required, the program repeats previous steps until a sufficient number of rules are generated.

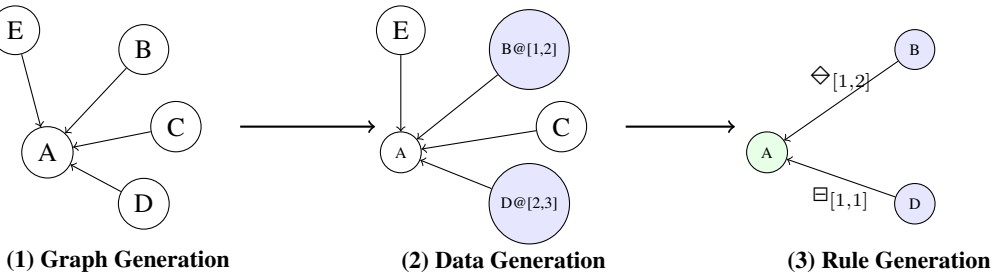

| (1) Graph Generation | (2) Data Generation | (3) Rule Generation |

Figure 3: An example of generating temporal data and rules. First, we randomly generate a graph. Next, our program selects specific nodes to assign time points. In our example, nodes B and D are chosen, resulting in two temporal facts: $\{B@[1,2], D@[2,3]\}$; Finally, we select a node as the head atom, with body atoms derived from the previous step. We then randomly assign temporal operators to these body atoms, resulting in the rule: $A \leftarrow \diamondsuit_{[1,2]}B \wedge \boxminus_{[1,1]}D$. The number of body atoms, the time range, and the temporal operators are specified as input parameters.

An example of generating temporal data and rules is shown in Figure 3. In particular, our program will have a post-processing operation to scan all the data and rules to ensure they have been utilized and removes any data and rules (in the ablation study, we will explore the impact of irrelevant data and rules) that are not participated in the the temporal reasoning process. We define the following flags for the samples to be generated based on their characteristics: `rational number`, `multiple body atoms`, `recursive` and `mixed operators`. These flags control the rule structures during the generation process.

|  | Prompt type | SingleAtom | MultiAtoms | Rational | MixedOperators | MultiRules | Recursive |
|---|---|---|---|---|---|---|---|
|  | Zero-shot | 45.8 | 43.2 | 37.1 | 57.3 | 53.3 | 37.7 |
| GPT-4o | Few-shot | 40.4 | 38.0 | 27.2 | 51.6 | 36.7 | 32.2 |
|  | Zero-shot-CoT | **85.6** | **85.1** | **85.7** | **90.3** | **74.0** | **58.0** |
|  | Zero-shot | 40.7 | 44.0 | 43.9 | 60.5 | 39.1 | 8.7 |
| Llama-3 | Few-shot | 38.4 | 44.3 | 44.4 | 47.1 | 36.1 | 30.2 |
|  | Zero-shot-CoT | **59.9** | **58.4** | **68.2** | **64.1** | **59.0** | **48.5** |

Table 1: Model performance on the synthetic benchmarks across six rule structures, as defined in Section 3.1, shows that GPT-4o with chain-of-thought prompting significantly outperforms its counterparts—zero-shot and few-shot—by approximately 30% to 50. This indicates that it is crucial for LLMs to generate the reasoning process before arriving at the final conclusion. Specifically, we observe that the three prompting settings for Llama-3 achieve similar performance, suggesting that Llama-3 struggles with temporal reasoning in symbolic forms.

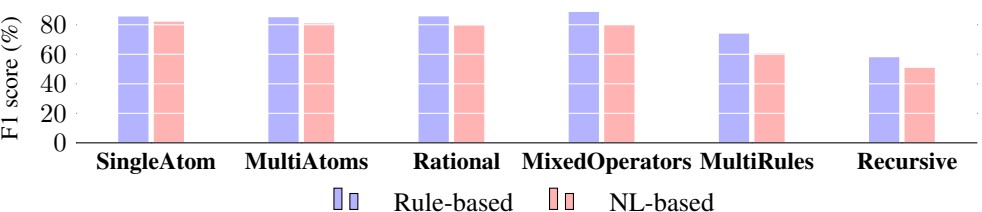

Figure 4: F1 score (%) on three benchmarks with *rule-based* and natural language (NL) based temporal logic reasoning and using Zero-shot-CoT.

## 4 EXPERIMENTS AND RESULTS

Based on our synthesized benchmarks, we aim to investigate *whether large language models can solve temporal reasoning problems from the traditional logic-based AI domain* by evaluating two large language models (GPT-4o[4] and Llama-3[5]) with three different kinds of prompting settings.

**Baselines** We adopt three basic prompting approaches as baselines. Specifically, zero-shot prompting (ZERO-SHOT), few-shot in-context learning (FEW-SHOT) (Brown et al., 2020), chain-of-thought prompting (COT) (Wei et al., 2022), and zero-shot chain-of-thought (0-COT) (Kojima et al., 2022) are leveraged to tackle various temporal reasoning problems in our constructed benchmark.

**Benchmark statistics and experimental settings** Unless otherwise specified, each benchmark level contains 200 samples selected from the facts derived using the chosen data and rule(s). For negative samples, a random interval is chosen, ensuring that these intervals do not overlap with those of the derived facts. Specifically, for all baselines, the temperature value is set to 0. For few-shot prompting techniques, the input prompt includes two manually constructed exemplars. In this paper, we use the F1 score as the evaluation metric. Details about the benchmark statistics, instructions, exemplars, and other relevant experimental information can be found in the the Appendix part.

### 4.1 MAIN RESULTS

We observe that for GPT-4o, the *zero-shot* and *few-shot* prompting settings achieve extremely low accuracy across the six benchmarks, with some experiencing more than a 50% accuracy drop compared to *zero-shot-CoT*. This indicates the necessity of using the chain-of-thought strategy with LLMs to solve temporal reasoning problems effectively. Notably, on the three benchmarks **SingleAtom**, **MultiAtoms**, and **Rational**, we observe similar accuracy with less than 2% variance between them. Interestingly, we found that using multiple operators yields the best performance. This is surprising because we assumed that employing more operators would require the model to understand more semantics of DataloMTL, thus increasing the problem's complexity compared to using only one type of MTL operator. However, considering the reasoning process using the symbolic reasoner, such results are also reasonable because if two rules contain the same number of body atoms, the number of calculating operators is the same. From this perspective, LLMs exhibit similar behavior to symbolic reasoner-based approaches.

In the **MultiRules** and **Recursive** benchmarks, we notice a significant performance drop, with **Recursive** achieving the lowest performance at 64.5%. These results suggest that recursive rules are particularly challenging for models, as they require not only understanding the language semantics and performing step-by-step reasoning but also possessing strong *inductive ability*. Unfortunately, the results achieved by Llama-3 are almost random, indicating that the model lacks the capability to solve symbolic form temporal reasoning problems. One possible reason is the model size being too small, preventing it from generalizing to unseen tasks. Additionally, the lack of relevant training datasets during pre-training or fine-tuning stages could also contribute to this issue.

---

[4] https://openai.com/index/hello-gpt-4o/
[5] https://huggingface.co/meta-llama/Meta-Llama-3-8B-Instruct

## 4.2 RULE BASED VS. NATURAL LANGUAGE (NL) BASED TEMPORAL LOGIC REASONING

In this paper, our primary focus is on benchmarking and testing the temporal logic reasoning capabilities of LLMs by evaluating them in a rule-based format. However, it is also interesting to compare this with another setting: verbalizing the rules before presenting them to the LLMs, as explored in many previous similar works like CronQuestions Saxena et al. (2021); Ismayilzada et al. (2023). Given that manually converting each rule into its corresponding natural language expression is a labor-intensive process, we adopt a template-based approach to automate this verbalization. Although this method may result in expressions that are not entirely natural[6], it provides a practical alternative to manual translation.

From Figure 4, we observe that both the rule-based and natural language-based settings achieve similar results, with the rule-based approach performing slightly better. The comparison indicates that LLMs are also capable of understanding the semantics of input expressed in rules, provided that each notation is clearly explained in the instructions. Notably, both settings struggle with the **MultiRules** and **Recursive** cases. One possible reason for this is that, while LLMs can understand the semantics of temporal logic language, they still face significant challenges in executing multiple deductions, retaining intermediate results, and recognizing repeated patterns—tasks that require delicate algorithms to accomplish effectively in the field of temporal logic Wałęga et al. (2023a).

## 4.3 ABLATION STUDY

To explore which component of the rule structure most significantly impact the reasoning complexity for LLMs, we designed four sets of ablation study experiments. These experiments explored the effects of the number of relevant rules , the number of operators considered, the percentage of irrelevant data, and the percentage of irrelevant rules. From Figure 5 (a), we observe that as the lengths of dependent rules increase, the model's performance noticeably degrades. One possible reason is that when multiple rules are mutually dependent, the model needs to store intermediate results during the derivation process to complete subsequent steps that rely on previously derived outcomes. Unlike symbolic reasoners, which can explicitly store intermediate results, it may be challenging for large language models (LLMs) to retain such information in an auto-regressive manner. Additionally, Figure 5 (b) demonstrates that using more types of operators does not affect reasoning complexity, indicating that understanding the semantics of the temporal logic language is not a major issue for the model. Lastly, the results in Figures 5 (c) and (d) show that the model's performance is only minimally affected by irrelevant information, whether in the form of temporal data or rules, demonstrating its ability to correctly select relevant rules and remain resistant to distracting information.

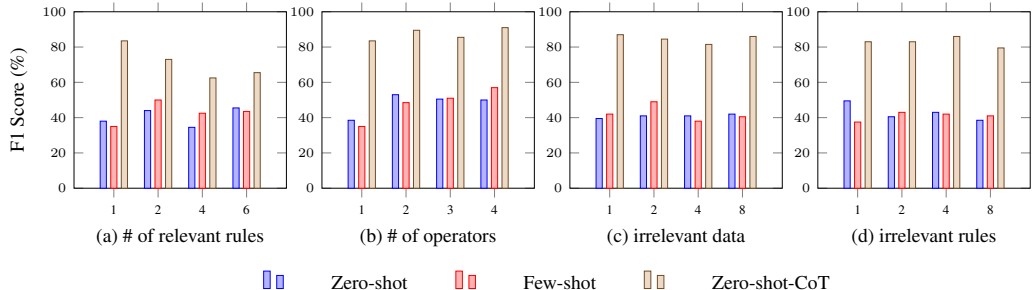

Figure 5: Results of ablation study. (a) presents the benchmark results for temporal rules of varying lengths; (b) illustrates the benchmark results for different operators; (c) displays the benchmark results with varying percentages of irrelevant data relative to relevant data (e.g., if there are 10 temporal facts required for the temporal reasoning process, 100% denotes that an additional 10 irrelevant temporal facts are included in the input); (d) showcases the benchmark results with varying percentages of irrelevant rules relative to target rules (the meaning of percentages in the x-axis is similar to the setting of adding irrelevant temporal facts.

---

[6]Some examples of these conversions can be found in the Appendix.

## 5 RELATED WORKS

**Symbolic approaches for temporal reasoning**   Knowledge representation languages, such as Linear Temporal Logic (LTL) Huth & Ryan (2004) and DatalogMTL Brandt et al. (2018), have become the de facto standard for specifying temporal properties in both formal verification and artificial intelligence. Many temporal reasoning problems have proven to be PSPACE-complete Wałęga et al. (2019); Fionda & Greco (2018); Bauland et al. (2009); Wałęga et al. (2020). *Satisfiability checking*, that is, the problem of deciding whether a given formula admits a satisfying model, is one of the most important computational tasks associated with the logic, and one of the first that have been carefully studied Sistla & Clarke (1985). Similarly, the main reasoning tasks considered in DatalogMTL are *fact entailment* and *consistency checking*. These problems polynomially reduce to the complements of each other Brandt et al. (2018). Despite this theoretically high computational complexity, numerous techniques and tools have been developed to solve different temporal reasoning problems, which range from tableau systems Goré & Widmann (2009); Bertello et al. (2016) to reductions to model checking Cavada et al. (2014), to automata techniques Li et al. (2014); Wang et al. (2022).

**Prompting LLMs for temporal reasoning**   Although the aforementioned temporal reasoning problems have been widely explored in the traditional logic-based AI domain, they remain underexplored in the regime of LLMs. There is a substantial body of research evaluating the reasoning abilities of LLMs in an in-context learning setting, covering areas such as arithmetic reasoning, logical reasoning, and commonsense reasoning. Notably, simple math problem datasets like AQUA Ling et al. (2017), GSM8K (Cobbe et al., 2021), and SVAMP (Patel et al., 2021) are frequently used to assess arithmetic reasoning (Touvron et al., 2023; Shi et al., 2023). Welleck et al. (2021) developed NaturalProofs, a multi-domain dataset for studying mathematical reasoning in natural language, while Welleck et al. (2022) investigated LLMs' abilities to generate the next step in mathematical proofs and complete full proofs. Additionally, LLMs have been evaluated on logical reasoning tasks, including symbolic tasks like Coin Flip and Last Letter Concatenation (Wei et al., 2022), and Logic Grid Puzzles on the BIG-BENCH (Srivastava et al., 2023). Commonsense reasoning datasets (Talmor et al., 2019) have also been proposed for evaluating LLMs. Most relevant to our work are various approaches to evaluating and enhancing the algorithmic reasoning abilities of LLMs (Zhou et al., 2022). In this work, we focus on evaluating LLMs on temporal logic reasoning tasks.

## 6 CONCLUSION

We present TBEN, a new synthetic benchmark consisting of rule-based temporal logic reasoning problems, designed to test the temporal reasoning abilities of LLMs. TBEN contains diverse temporal reasoning problems of varying complexities, providing flexible configurations for customising generated benchmarks. Our experimental results reveal that models have difficulties in understanding symbolic data and performing temporal reasoning. Even though powerful LLMs like GPT-4o have been trained on extensive data from across the Internet, they struggle with some simple temporal reasoning tasks. By proposing this benchmark, we aim to draw attention to the data-based temporal reasoning abilities of LLMs. Our goal is to facilitate the development of specialized methods to enhance these abilities, such as training models in the agent reasoning style with data as input and equipping models with more temporal reasoning and symbolic knowledge.

## 7 LIMITATION

Our experiments were constrained by the speed, computational resources, and financial costs associated with utilizing the closed-source GPT-4o. For instance, although our generator allows for the creation of benchmarks of temporal data and rules with arbitrary sizes, we obtained results across multiple temporal reasoning datasets of varying complexities on a relatively small scale due to the financial costs associated with GPT-4o API calls.

Another limitation of this preliminary exploration into testing the temporal reasoning abilities of LLMs is that we only present experimental results from three prompting settings, despite the availability of many other advanced prompting strategies. Furthermore, while we demonstrate that a chain-of-thought approach is necessary for solving temporal reasoning problems, we do not offer an effective method for enhancing the LLM's ability to handle temporal logic reasoning problems.

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

## A    PROMPTS USED IN THE BASIC EVALUATION

For all evaluations, we prepend a system message to introduce the syntax of DatalogMTL language as below:

---

You are given a dataset and a temporal rule, and your task is to judge whether the given fact is entailed by the dataset and the rule.

The rules are expressed as DatalogMTL, a knowledge representation language that extends Datalog with operators from metric temporal logic (MTL). The semantics of four MTL operators are given as follows:

If Diamondminus[a,b]A is true at the time t, it requires that A needs to be true at some time between t-b and t-a.

If Boxminus[a,b]A is true at the time t, it requires that A needs to be true continuously between t-b and t-a.

If Diamondplus[a,b]A is true at the time t, it requires that A needs to be true at some point between t+a and t+b.

If Boxplus[a,b]A is true at the time t, it requires that A needs to be true continuously between t+a and t+b.

---

**Zero-shot**    For zero-shot evaluations, the system prompt we uses is the above general introduction plus the statement: *You should not give any explanation and you should only output "true" or "false".* We are using the statement *Now we have some temporal data and some rules, data: {data} rule: {rule}, Is {inquiry} true or not?* as the user prompt to evaluate LLM's reasoning ability.

Here is an example of the complete prompt we constructed to do zero-shot evaluation.

| System Prompt | You are given a dataset and a temporal rule, and your task is to judge whether the given fact is entailed by the dataset and the rule. |
|---|---|
| | The rules are expressed as DatalogMTL, a knowledge representation language that extends Datalog with operators from metric temporal logic (MTL). The semantics of four MTL operators are given as follows: |
| | If Diamondminus[a,b]A is true at the time t, it requires that A needs to be true at some time between t-b and t-a. |
| | If Boxminus[a,b]A is true at the time t, it requires that A needs to be true continuously between t-b and t-a. |
| | If Diamondplus[a,b]A is true at the time t, it requires that A needs to be true at some point between t+a and t+b. |
| | If Boxplus[a,b]A is true at the time t, it requires that A needs to be true continuously between t+a and t+b. |
| | You should not give any explanation and you should only output "true" or "false" |
| User Prompt | Now we have some temporal data and some rules, data: B@[3,10] rule: A:-Diamondplus[6,10]B Is A@[1,4] true or not? |
| LLM's output | false |
| Expected Answer | true |

**Few-shot**    For few-shot evaluations, just like the zero-shot case, the system prompt we uses is the above general introduction plus the statement: *You should not give any explanation and you should only output "true" or "false".* However, in the user prompt, we are integrating some examples using the following syntax:

> To help you better understand the task, I will provide two examples.
> Example 1: data: {pos data} rule: {pos rule} in this case you should output "true" for {pos inquiry}.
> Example 2: data: {neg data} rule: {neg rule} in this case you should output "false" for {neg inquiry}.
> Now we have some temporal data and some rules, data: {data} rule: {rule}
> Is {inquiry} true or not?"

*{pos data}*, *{pos rule}* and *{pos inquiry}* are from a positive sample, *{neg data}*, *{neg rule}* and *{neg inquiry}* are from a negative sample. They are samples not in the testing set, but has the same type as the testing samples.

Here is an example of the complete prompt we constructed to do few-shot evaluation.

| System Prompt | You are given a dataset and a temporal rule, and your task is to judge whether the given fact is entailed by the dataset and the rule. |
| --- | --- |
| | The rules are expressed as DatalogMTL, a knowledge representation language that extends Datalog with operators from metric temporal logic (MTL). The semantics of four MTL operators are given as follows: |
| | If Diamondminus[a,b]A is true at the time t, it requires that A needs to be true at some time between t-b and t-a. |
| | If Boxminus[a,b]A is true at the time t, it requires that A needs to be true continuously between t-b and t-a. |
| | If Diamondplus[a,b]A is true at the time t, it requires that A needs to be true at some point between t+a and t+b. |
| | If Boxplus[a,b]A is true at the time t, it requires that A needs to be true continuously between t+a and t+b. |
| | You should not give any explanation and you should only output "true" or "false" |
| User Prompt | To help you better understand the task, I will provide two examples. |
| | Example 1: data: B@[5,7] |
| | rule: A:-Boxminus[10,12]B |
| | in this case you should output "true" for A@[17,17] |
| | Example 2: data: B@[1,9] |
| | rule: A:-Diamondplus[3,3]B |
| | in this case you should output "false" for A@[-25,-6] |
| | Now we have some temporal data and some rules, data: B@[3,10] |
| | rule: A:-Diamondplus[6,10]B |
| | Is A@[1,4] true or not?" |
| LLM's output | false |
| Expected Answer | true |

**Zero-shot-CoT**  For zero-shot-cot evaluations, the system prompt we uses is the above general introduction without the additional the statement that we used in zero-shot or few-shot to ask LLM's to answer directly. We are going to put some directions to let LLM think step by step in the user prompt below:

> Now we have some temporal data and some rules, data: {data}
> rule: {rule}
> Is {inquiry} true or not? Do not answer directly, think step by step.

After that, we will put the LLM's response in the context and then asks LLM again about the final answer by *Based on your previous response, without any explanation state your answers to the question. You should only output "true" or "false".*

Here is an example of the complete prompt we constructed to do zero-shot evaluation.

| | |
|---|---|
| System Prompt | You are given a dataset and a temporal rule, and your task is to judge whether the given fact is entailed by the dataset and the rule.
The rules are expressed as DatalogMTL, a knowledge representation language that extends Datalog with operators from metric temporal logic (MTL). The semantics of four MTL operators are given as follows:
If Diamondminus[a,b]A is true at the time t, it requires that A needs to be true at some time between t-b and t-a.
If Boxminus[a,b]A is true at the time t, it requires that A needs to be true continuously between t-b and t-a.
If Diamondplus[a,b]A is true at the time t, it requires that A needs to be true at some point between t+a and t+b.
If Boxplus[a,b]A is true at the time t, it requires that A needs to be true continuously between t+a and t+b. |
| User Prompt | Now we have some temporal data and some rules, data: B@[3,10]
rule: A:-Diamondplus[6,10]B
Is A@[1,4] true or not? Do not answer directly, think step by step. |
| LLM's output | First, let's understand the rule: A:-Diamondplus[6,10]B. This rule means that if B is true at some point between t+6 and t+10, then A is true at time t.
Now, let's look at the data: B@[3,10]. This means that B is true at some point between time 3 and time 10.
We are asked to determine if A@[1,4] is true or not. This means we are looking to see if A is true at some point between time 1 and time 4.
Given the rule, for A to be true at time 1, B would need to be true at some point between time 7 (1+6) and 11 (1+10). For A to be true at time 4, B would need to be true at some point between time 10 (4+6) and 14 (4+10). However, B is only true between time 3 and time 10. Therefore, B is not true at any point between time 7 and 11 or between time 10 and 14.
So, A@[1,4] is not true. |
| User Prompt | Based on your previous response, without any explanation state your answers to the question. You should only output "true" or "false" |
| LLM's output | false |
| Expected Answer | true |

## B  PROMPTS USED IN THE NATURAL LANGUAGE EVALUATION

For the evaluations using natural language, we no longer need to introduce the syntax of DatalogMTL language. Therefore there is no general system prompt.

**Zero-shot**   For zero-shot setting, we change the system prompt to a simple sentence to ensure that LLMs output in a desired format *You should not give any explanation and you should only output "true" or "false"*. We are using the statement *Now we have some temporal data and some rules, data: {data} rule: {rule}, Is {inquiry} true or not?* as the user prompt to evaluate LLM's reasoning ability. {data}, {rule} and {inquiry} are all replaced by their verbalized representation.

Here is an example of the complete prompt we constructed to do zero-shot evaluation.

| | |
|---|---|
| System Prompt | You should not give any explanation and you should only output "true" or "false" |
| User Prompt | Now we have some temporal data and some rules, data:
A holds From 10.000 to 10.000
rule: B holds in each time such that A will hold sometime between 4.000 and 15.000 hours in the future
Is B holds From -5.000 to 1.000 true or not? |
| LLM's output | false |
| Expected Answer | true |

**Few-shot** For few-shot evaluations, just like the zero-shot case, the system prompt we uses is the same: *You should not give any explanation and you should only output "true" or "false"*. However, in the user prompt, we are integrating some examples using the following syntax:

---

To help you better understand the task, I will provide two examples.
Example 1: data: {pos data} rule: {pos rule} in this case you should output "true" for {pos inquiry}.
Example 2: data: {neg data} rule: {neg rule} in this case you should output "false" for {neg inquiry}.
Now we have some temporal data and some rules, data: {data} rule: {rule}

---

*{pos data}*, *{pos rule}* and *{pos inquiry}* are verbalized representations from a positive sample, *{neg data}*, *{neg rule}* and *{neg inquiry}* are verbalized representations from a negative sample. They are samples not in the testing set, but has the same type as the testing samples.

Here is an example of the complete prompt we constructed to do few-shot evaluation.

| System Prompt | You should not give any explanation and you should only output "true" or "false" |
|---|---|
| User Prompt | To help you better understand the task, I will provide two examples. Example 1: data: B holds From 5.000 to 7.000 rule: A holds in each time such that B did hold continuously for 10.000 and 12.000 hours before in this case you should output "true" for A holds From 17.000 to 17.000 Example 2: data: B holds From 1.000 to 9.000 rule: A holds in each time such that B will hold sometime between 3.000 and 3.000 hours in the future in this case you should output "false" for A holds From -25.000 to -6.000 Now we have some temporal data and some rules, data: B holds From 3.000 to 10.000 rule: A holds in each time such that B will hold sometime between 6.000 and 10.000 hours in the future Is A holds From 1.000 to 4.000 true or not? |
| LLM's output | true |
| Expected Answer | true |

**Zero-shot-CoT** For verbalized zero-shot-cot evaluations, we no longer use system message since we don't need to introduce the synatx nor ask it to answer in a specific format. We organize our user prompts in the following format:

---

Now we have some temporal data and some rules, data: {data}
rule: {rule}
Is {inquiry} true or not? Do not answer directly, think step by step.

---

where {data}, {rule} and {inquiry} refers to the verbalized rules.

After that, we will put the LLM's response in the context and then asks LLM again about the final answer by *Based on your previous response, without any explanation state your answers to the question. You should only output "true" or "false".*

Here is an example of the complete verbalized prompt we constructed to do zero-shot evaluation.

| | |
|---|---|
| System Prompt | |
| User Prompt | Now we have some temporal data and some rules, data: 
 B holds From 3.000 to 10.000 
 rule: A holds in each time such that B will hold sometime between 6.000 and 10.000 hours in the future 
 Is A holds From 1.000 to 4.000 true or not? Do not answer directly, think step by step. |
| LLM's output | To determine if A holds from 1.000 to 4.000 based on the given rule and data, let's break it down step by step: 
 1. **Understand the Data:** 
 - B holds from 3.000 to 10.000. 
 2. **Understand the Rule:** 
 - A holds at each time such that B will hold sometime between 6.000 and 10.000 hours in the future. 
 3. **Analyze the Time Intervals:** 
 - We need to check if B holds in the interval [6.000, 10.000] hours in the future from any time within [1.000, 4.000]. 
 4. **Check Each Time Point in [1.000, 4.000]:** 
 - For time 1.000: 
 - 6.000 to 10.000 hours in the future from 1.000 is [7.000, 11.000]. 
 - B holds from 3.000 to 10.000, which overlaps with [7.000, 10.000]. 
 - Therefore, B holds sometime in [7.000, 11.000]. 
 - For time 2.000: 
 - 6.000 to 10.000 hours in the future from 2.000 is [8.000, 12.000]. 
 - B holds from 3.000 to 10.000, which overlaps with [8.000, 10.000]. 
 - Therefore, B holds sometime in [8.000, 12.000]. 
 - For time 3.000: 
 - 6.000 to 10.000 hours in the future from 3.000 is [9.000, 13.000]. 
 - B holds from 3.000 to 10.000, which overlaps with [9.000, 10.000]. 
 - Therefore, B holds sometime in [9.000, 13.000]. 
 - For time 4.000: 
 - 6.000 to 10.000 hours in the future from 4.000 is [10.000, 14.000]. 
 - B holds from 3.000 to 10.000, which overlaps with [10.000, 10.000]. 
 - Therefore, B holds sometime in [10.000, 14.000]. 
 5. **Conclusion:** 
 - For each time point in the interval [1.000, 4.000], B holds sometime in the future interval [6.000, 10.000] hours from that time point. 
 - Therefore, according to the rule, A holds from 1.000 to 4.000. 
 So, based on the given rule and data, A holds from 1.000 to 4.000. |
| User Prompt | Based on your previous response, without any explanation state your answers to the question. You should only output "true" or "false" |
| LLM's output | true |
| Expected Answer | true |

## C   DETAILED INFORMATION OF OUR BENCHMARK

We generated different number of samples based on the type of DatalogMTl, such as SingleAtom, MultiAtoms, Rational, MixedOperators, MultiRules and Recursive.

| Type of Sample | # of Positive Samples | # of Negative Samples |
|---|---|---|
| SingleAtom | 500 | 500 |
| MultiAtoms | 300 | 300 |
| Rational | 500 | 500 |
| MixedOperators (with 2 operators) | 1739 | 1739 |
| MixedOperators (with 3 operators) | 145 | 145 |
| MixedOperators (with 3 operators) | 126 | 126 |
| MultiRules (with 2 rules) | 250 | 250 |
| MultiRules (with 4 rules) | 250 | 250 |
| MultiRules (with 6 rules) | 150 | 150 |
| Recursive | 500 | 500 |

Table 2: The number of samples of different categories in our dataset

No matter how many samples are there in the our dataset, for each category, we only use the first 100 positive samples and the last 100 negative samples as the testing samples. For in-context/few-shot evaluations, we use one additional positive sample (the 101-th sample) and one additional negative sample (the last 101-th sample) as in-context learning sample to put into the context.

For MultiAtoms, we don't specify the number of operators it has in the rule nor evaluate them separately, while in general it follows the following distribution:

| Type of Sample | # of Positive Samples | # of Negative Samples |
|---|---|---|
| MultiAtoms (with 2 atoms in the rule) | 109 | 115 |
| MultiAtoms (with 3 atoms in the rule) | 79 | 79 |
| MultiAtoms (with 4 atoms in the rule) | 61 | 64 |
| MultiAtoms (with 5 atoms in the rule) | 51 | 42 |
| Total | 300 | 300 |

Table 3: The distribution of the number of atoms in our MultiAtoms subset of our dataset

## D  BAD CASE ANALYSIS

### D.1  CASES THAT ARE INCORRECT IN SYMBOLIZED REPRESENTATION BUT ARE CORRECT IN VERBALIZED REPRESENTATION

| | Symbolic | Verbalized |
|---|---|---|

| System Prompt | You are given a dataset and a temporal rule, and your task is to judge whether the given fact is entailed by the dataset and the rule. The rules are expressed as DatalogMTL, a knowledge representation language that extends Datalog with operators from metric temporal logic (MTL). The semantics of four MTL operators are given as follows: If Diamondminus[a,b]A is true at the time t, it requires that A needs to be true at some time between t-b and t-a. If Boxminus[a,b]A is true at the time t, it requires that A needs to be true continuously between t-b and t-a. If Diamondplus[a,b]A is true at the time t, it requires that A needs to be true at some point between t+a and t+b. If Boxplus[a,b]A is true at the time t, it requires that A needs to be true continuously between t+a and t+b. | (No introduction to the syntax is required for verbalized representations) |
|---|---|---|
| User Prompt | Now we have some temporal data and some rules, data: B@[3,10] rule: A:-Diamondplus[6,10]B Is A@[1,4] true or not? Do not answer directly, think step by step. | Now we have some temporal data and some rules, data: B holds From 3.000 to 10.000 rule: A holds in each time such that B will hold sometime between 6.000 and 10.000 hours in the future Is A holds From 1.000 to 4.000 true or not? Do not answer directly, think step by step. |

| LLM's output | First, let's understand the rule: A:-Diamondplus[6,10]B. This rule means that if B is true at some point between t+6 and t+10, then A is true at time t. 

 Now, let's look at the data: B@[3,10]. This means that B is true at some point between time 3 and time 10. 

 We are asked to determine if A@[1,4] is true or not. This means we are looking to see if A is true at some point between time 1 and time 4. 

 Given the rule, for A to be true at time 1, B would need to be true at some point between time 7 (1+6) and 11 (1+10). For A to be true at time 4, B would need to be true at some point between time 10 (4+6) and 14 (4+10). 

 However, B is only true between time 3 and time 10. Therefore, B is not true at any point between time 7 and 11 or between time 10 and 14. So, A@[1,4] is not true. | To determine if A holds from 1.000 to 4.000 based on the given rule and data, let's break it down step by step: 
 1. **Understand the Data:** 
 - B holds from 3.000 to 10.000. 
 2. **Understand the Rule:** 
 - A holds at each time such that B will hold sometime between 6.000 and 10.000 hours in the future. 
 3. **Analyze the Time Intervals:** 
 - We need to check if B holds in the interval [6.000, 10.000] hours in the future from any time within [1.000, 4.000]. 
 4. **Check Each Time Point in [1.000, 4.000]:** 
 - For time 1.000: 
 - 6.000 to 10.000 hours in the future from 1.000 is [7.000, 11.000]. 
 - B holds from 3.000 to 10.000, which overlaps with [7.000, 10.000]. 
 - Therefore, B holds sometime in [7.000, 11.000]. 
 - For time 2.000: 
 - 6.000 to 10.000 hours in the future from 2.000 is [8.000, 12.000]. 
 - B holds from 3.000 to 10.000, which overlaps with [8.000, 10.000]. 
 - Therefore, B holds sometime in [8.000, 12.000]. 
 - For time 3.000: 
 - 6.000 to 10.000 hours in the future from 3.000 is [9.000, 13.000]. 
 - B holds from 3.000 to 10.000, which overlaps with [9.000, 10.000]. 
 - Therefore, B holds sometime in [9.000, 13.000]. 
 - For time 4.000: 
 - 6.000 to 10.000 hours in the future from 4.000 is [10.000, 14.000]. 
 - B holds from 3.000 to 10.000, which overlaps with [10.000, 10.000]. 
 - Therefore, B holds sometime in [10.000, 14.000]. 
 5. **Conclusion:** 
 - For each time point in the interval [1.000, 4.000], B holds sometime in the future interval [6.000, 10.000] hours from that time point. 
 - Therefore, according to the rule, A holds from 1.000 to 4.000. 
 So, based on the given rule and data, A holds from 1.000 to 4.000. |
|---|---|---|

| User Prompt | Based on your previous response, without any explanation state your answers to the question. You should only output "true" or "false" | |
|---|---|---|
| LLM's output | false | true |
| Expected Answer | true | |

## D.2 CASES THAT ARE INCORRECT IN ZERO-SHOT BY ARE CORRECT WHEN CHAIN OF THOUGHT IS APPLIED

Example 1:

| | Without CoT | With CoT |
|---|---|---|
| System Prompt | You are given a dataset and a temporal rule, and your task is to judge whether the given fact is entailed by the dataset and the rule.

The rules are expressed as DatalogMTL, a knowledge representation language that extends Datalog with operators from metric temporal logic (MTL). The semantics of four MTL operators are given as follows:
If Diamondminus[a,b]A is true at the time t, it requires that A needs to be true at some time between t-b and t-a.
If Boxminus[a,b]A is true at the time t, it requires that A needs to be true continuously between t-b and t-a.
If Diamondplus[a,b]A is true at the time t, it requires that A needs to be true at some point between t+a and t+b.
If Boxplus[a,b]A is true at the time t, it requires that A needs to be true continuously between t+a and t+b.
You should not give any explanation and you should only output "true" or "false" | You are given a dataset and a temporal rule, and your task is to judge whether the given fact is entailed by the dataset and the rule.

The rules are expressed as DatalogMTL, a knowledge representation language that extends Datalog with operators from metric temporal logic (MTL). The semantics of four MTL operators are given as follows:
If Diamondminus[a,b]A is true at the time t, it requires that A needs to be true at some time between t-b and t-a.
If Boxminus[a,b]A is true at the time t, it requires that A needs to be true continuously between t-b and t-a.
If Diamondplus[a,b]A is true at the time t, it requires that A needs to be true at some point between t+a and t+b.
If Boxplus[a,b]A is true at the time t, it requires that A needs to be true continuously between t+a and t+b. |
| User Prompt | Now we have some temporal data and some rules, data:
B@[6,9]
rule: A:-Diamondminus[6,15]B
Is A@[12,18] true or not? | Now we have some temporal data and some rules, data:
B@[6,9]
rule: A:-Diamondminus[6,15]B
Is A@[12,18] true or not? Do not answer directly, think step by step. |

| | | |
|---|---|---|
| LLM's output | | First, let's understand the rule: A:-Diamondminus[6,15]B. This rule means that if B is true at some point between t-15 and t-6, then A is true at time t.

Now, let's consider the fact A@[12,18]. This means we are checking if A is true at some point between time 12 and 18.

To check this, we need to see if B is true at some point between (12-15) and (12-6), i.e., between time -3 and 6, or between (18-15) and (18-6), i.e., between time 3 and 12.

From the given data, we know that B is true between time 6 and 9. This falls within the required time range for A to be true.

Therefore, A@[12,18] is true. |
| User Prompt | | Based on your previous response, without any explanation state your answers to the question. You should only output "true" or "false" |
| LLM's output | false | true |
| Expected Answer | true | |

Example 2:

| | Without CoT | With CoT |
|---|---|---|
| System Prompt | You are given a dataset and a temporal rule, and your task is to judge whether the given fact is entailed by the dataset and the rule.

The rules are expressed as DatalogMTL, a knowledge representation language that extends Datalog with operators from metric temporal logic (MTL). The semantics of four MTL operators are given as follows:
If Diamondminus[a,b]A is true at the time t, it requires that A needs to be true at some time between t-b and t-a.
If Boxminus[a,b]A is true at the time t, it requires that A needs to be true continuously between t-b and t-a.
If Diamondplus[a,b]A is true at the time t, it requires that A needs to be true at some point between t+a and t+b.
If Boxplus[a,b]A is true at the time t, it requires that A needs to be true continuously between t+a and t+b.
You should not give any explanation and you should only output "true" or "false" | You are given a dataset and a temporal rule, and your task is to judge whether the given fact is entailed by the dataset and the rule.

The rules are expressed as DatalogMTL, a knowledge representation language that extends Datalog with operators from metric temporal logic (MTL). The semantics of four MTL operators are given as follows:
If Diamondminus[a,b]A is true at the time t, it requires that A needs to be true at some time between t-b and t-a.
If Boxminus[a,b]A is true at the time t, it requires that A needs to be true continuously between t-b and t-a.
If Diamondplus[a,b]A is true at the time t, it requires that A needs to be true at some point between t+a and t+b.
If Boxplus[a,b]A is true at the time t, it requires that A needs to be true continuously between t+a and t+b. |

| | | |
|---|---|---|
| User Prompt | Now we have some temporal data and some rules, data:
A@[8,13]
rule: B:-Diamondminus[10,12]A
Is B@[21,24] true or not? | Now we have some temporal data and some rules, data:
A@[8,13]
rule: B:-Diamondminus[10,12]A
Is B@[21,24] true or not? Do not answer directly, think step by step. |
| LLM's output | | First, let's understand the rule: B:-Diamondminus[10,12]A. This rule means that if A is true at some point between t-12 and t-10, then B is true at time t.
Now, let's look at the data: A@[8,13]. This means that A is true continuously from time 8 to time 13.
Now, we need to check if B@[21,24] is true or not. For B to be true at any time between 21 and 24, A needs to be true at some point between (t-12) and (t-10).
Let's check for each time t in [21,24]:
- For t=21, A needs to be true at some point between 21-12=9 and 21-10=11. A is true at these times according to the data.
- For t=22, A needs to be true at some point between 22-12=10 and 22-10=12. A is true at these times according to the data.
- For t=23, A needs to be true at some point between 23-12=11 and 23-10=13. A is true at these times according to the data.
- For t=24, A needs to be true at some point between 24-12=12 and 24-10=14. A is true at time 12, but not at time 14. However, the rule requires A to be true at some point between these times, not at all times, so this condition is satisfied.
Therefore, B@[21,24] is true. |
| User Prompt | | Based on your previous response, without any explanation state your answers to the question. You should only output "true" or "false" |
| LLM's output | false | true |
| Expected Answer | true | |

## D.3 SINGLEATOM ERROR CASES

False Negative Example:

| System Prompt | You are given a dataset and a temporal rule, and your task is to judge whether the given fact is entailed by the dataset and the rule. The rules are expressed as DatalogMTL, a knowledge representation language that extends Datalog with operators from metric temporal logic (MTL). The semantics of four MTL operators are given as follows:If Diamondminus[a,b]A is true at the time t, it requires that A needs to be true at some time between t-b and t-a.If Boxminus[a,b]A is true at the time t, it requires that A needs to be true continuously between t-b and t-a.If Diamondplus[a,b]A is true at the time t, it requires that A needs to be true at some point between t+a and t+b.If Boxplus[a,b]A is true at the time t, it requires that A needs to be true continuously between t+a and t+b. |
|---|---|
| User Prompt | Now we have some temporal data and some rules, data: A@[7,15] 
 rule: B:-Boxminus[3,5]A 
 Is B@[8,30] true or not? Do not answer directly, think step by step. |
| LLM's output | To determine whether ( B@[8,30] ) is true given the data and the rule, we need to follow these steps: 
 1. **Understand the Rule**: The rule is ( B leftarrow Boxminus[3,5]A ). This means that ( B ) is true at time ( t ) if ( A ) is true continuously in the interval ([t-5, t-3]). 
 2. **Check the Interval for ( B@[8,30] )**: We need to check if ( B ) is true for every time point in the interval ([8, 30]). For each time ( t ) in this interval, ( A ) must be true continuously in the interval ([t-5, t-3]). 
 3. **Analyze the Data**: The data given is ( A@[7,15] ), which means ( A ) is true continuously from time 7 to time 15. 
 4. **Check Each Time Point in ([8, 30])**: 
 - For ( t = 8 ): The interval ([t-5, t-3]) is ([3, 5]). ( A ) is not true in this interval because ( A ) is only true from 7 to 15. 
 - For ( t = 9 ): The interval ([t-5, t-3]) is ([4, 6]). ( A ) is not true in this interval. 
 - For ( t = 10 ): The interval ([t-5, t-3]) is ([5, 7]). ( A ) is not true continuously in this interval (it starts at 7). 
 - For ( t = 11 ): The interval ([t-5, t-3]) is ([6, 8]). ( A ) is true continuously in this interval. 
 - For ( t = 12 ): The interval ([t-5, t-3]) is ([7, 9]). ( A ) is true continuously in this interval. 
 - For ( t = 13 ): The interval ([t-5, t-3]) is ([8, 10]). ( A ) is true continuously in this interval. 
 - For ( t = 14 ): The interval ([t-5, t-3]) is ([9, 11]). ( A ) is true continuously in this interval. 
 - For ( t = 15 ): The interval ([t-5, t-3]) is ([10, 12]). ( A ) is true continuously in this interval. 
 - For ( t = 16 ): The interval ([t-5, t-3]) is ([11, 13]). ( A ) is true continuously in this interval. 
 - For ( t = 17 ): The interval ([t-5, t-3]) is ([12, 14]). ( A ) is true continuously in this interval. 
 - For ( t = 18 ): The interval ([t-5, t-3]) is ([13, 15]). ( A ) is true continuously in this interval. 
 - For ( t = 19 ): The interval ([t-5, t-3]) is ([14, 16]). ( A ) is not true in this interval (it ends at 15). 
 - For ( t = 20 ) to ( t = 30 ): The intervals ([t-5, t-3]) will be ([15, 17]) to ([25, 27]). ( A ) is not true in any of these intervals. 
 5. **Conclusion**: ( B ) is not true for all time points in the interval ([8, 30]). Specifically, ( B ) is not true at ( t = 8, 9, 10, 19, ) and any ( t ) from 20 to 30. Therefore, ( B@[8,30] ) is not true. |
| User Prompt | Based on your previous response, without any explanation state your answers to the question. You should only output "true" or "false" |
| LLM's output | false |
| Expected Answer | true |

False Positive Example:

| System Prompt | You are given a dataset and a temporal rule, and your task is to judge whether the given fact is entailed by the dataset and the rule. The rules are expressed as DatalogMTL, a knowledge representation language that extends Datalog with operators from metric temporal logic (MTL). The semantics of four MTL operators are given as follows:If Diamondminus[a,b]A is true at the time t, it requires that A needs to be true at some time between t-b and t-a.If Boxminus[a,b]A is true at the time t, it requires that A needs to be true continuously between t-b and t-a.If Diamondplus[a,b]A is true at the time t, it requires that A needs to be true at some point between t+a and t+b.If Boxplus[a,b]A is true at the time t, it requires that A needs to be true continuously between t+a and t+b. |
|---|---|
| User Prompt | Now we have some temporal data and some rules, data: A@[0,3] rule: B:-Diamondminus[2,4]A Is B@[-34,8] true or not? Do not answer directly, think step by step. |
| LLM's output | To determine whether ( B@[-34,8] ) is true given the data and the rule, we need to follow these steps: 1. **Understand the Rule**: The rule is ( B leftarrow Diamondminus[2,4]A ). This means that ( B ) is true at time ( t ) if ( A ) is true at some time between ( t-4 ) and ( t-2 ). 2. **Determine the Time Interval for ( B@[-34,8] )**: We need to check if ( B ) is true at any time within the interval ([-34, 8]). 3. **Apply the Rule to Each Time Point in the Interval**: - For each time ( t ) in the interval ([-34, 8]), we need to check if ( A ) is true at some time between ( t-4 ) and ( t-2 ). 4. **Check the Data**: The data given is ( A@[0,3] ), which means ( A ) is true from time 0 to time 3. 5. **Evaluate the Rule for Each Time Point**: - For ( t = -34 ): We need ( A ) to be true between ([-38, -36]). Since ( A ) is only true between ([0, 3]), ( A ) is not true in this interval. - For ( t = -33 ): We need ( A ) to be true between ([-37, -35]). Again, ( A ) is not true in this interval. - Continue this process up to ( t = 8 ): - For ( t = 0 ): We need ( A ) to be true between ([-4, -2]). ( A ) is not true in this interval. - For ( t = 1 ): We need ( A ) to be true between ([-3, -1]). ( A ) is not true in this interval. - For ( t = 2 ): We need ( A ) to be true between ([-2, 0]). ( A ) is not true in this interval. - For ( t = 3 ): We need ( A ) to be true between ([-1, 1]). ( A ) is true at time 0, which is within this interval. - For ( t = 4 ): We need ( A ) to be true between ([0, 2]). ( A ) is true in this interval. - For ( t = 5 ): We need ( A ) to be true between ([1, 3]). ( A ) is true in this interval. - For ( t = 6 ): We need ( A ) to be true between ([2, 4]). ( A ) is true at time 2 and 3, which is within this interval. - For ( t = 7 ): We need ( A ) to be true between ([3, 5]). ( A ) is true at time 3, which is within this interval. - For ( t = 8 ): We need ( A ) to be true between ([4, 6]). ( A ) is not true in this interval. 6. **Conclusion**: Since ( A ) is true at some time between ( t-4 ) and ( t-2 ) for ( t = 3, 4, 5, 6, ) and ( 7 ), ( B ) is true at these times. Therefore, ( B ) is true at some points within the interval ([-34, 8]). Thus, ( B@[-34,8] ) is true. |
| User Prompt | Based on your previous response, without any explanation state your answers to the question. You should only output "true" or "false" |
| LLM's output | true |
| Expected Answer | false |

## D.4    RECURSIVE ERROR CASES

False Negative Example:

| System Prompt | You are given a dataset and a temporal rule, and your task is to judge whether the given fact is entailed by the dataset and the rule. The rules are expressed as DatalogMTL, a knowledge representation language that extends Datalog with operators from metric temporal logic (MTL). The semantics of four MTL operators are given as follows:If Diamondminus[a,b]A is true at the time t, it requires that A needs to be true at some time between t-b and t-a.If Boxminus[a,b]A is true at the time t, it requires that A needs to be true continuously between t-b and t-a.If Diamondplus[a,b]A is true at the time t, it requires that A needs to be true at some point between t+a and t+b.If Boxplus[a,b]A is true at the time t, it requires that A needs to be true continuously between t+a and t+b. |
|---|---|
| User Prompt | Now we have some temporal data and some rules, data: A@[10,12] rule: A:-Diamondplus[4,13]A Is A@[-469,-221] true or not? Do not answer directly, think step by step. |
| LLM's output | To determine whether ( A@[-469,-221] ) is true based on the given data and rule, we need to follow a step-by-step reasoning process. Let's break it down: ### Step 1: Understand the Data The data given is: [ A@[10,12] ] This means that ( A ) is true continuously from time 10 to time 12. ### Step 2: Understand the Rule The rule provided is: [ A leftarrow Diamondplus[4,13]A ] This rule states that ( A ) is true at time ( t ) if ( A ) is true at some point between ( t+4 ) and ( t+13 ). ### Step 3: Apply the Rule We need to check if ( A@[-469,-221] ) can be derived using the rule and the given data. #### Step 3.1: Check the Time Interval The interval ([-469, -221]) is quite far from the interval ([10, 12]) where ( A ) is known to be true. However, we need to see if the rule can propagate ( A ) backward or forward in time to cover the interval ([-469, -221]). #### Step 3.2: Propagate the Rule Forward Let's start from the known interval ([10, 12]) and see if we can propagate ( A ) forward using the rule. - For ( A ) to be true at time ( t ), ( A ) must be true at some point between ( t+4 ) and ( t+13 ). Starting from ( t = 10 ): - ( A ) is true at ( t = 10 ) if ( A ) is true at some point between ( 10+4 ) and ( 10+13 ), i.e., between ( 14 ) and ( 23 ). Starting from ( t = 12 ): - ( A ) is true at ( t = 12 ) if ( A ) is true at some point between ( 12+4 ) and ( 12+13 ), i.e., between ( 16 ) and ( 25 ). Since we do not have any data indicating that ( A ) is true in these new intervals, we cannot propagate ( A ) forward from the interval ([10, 12]). #### Step 3.3: Propagate the Rule Backward Now, let's consider if we can propagate ( A ) backward to cover the interval ([-469, -221]). - For ( A ) to be true at time ( t ), ( A ) must be true at some point between ( t+4 ) and ( t+13 ). To cover the interval ([-469, -221]), we would need to find a time ( t ) such that ( t+4 ) to ( t+13 ) overlaps with ([10, 12]). However, since ([10, 12]) is far in the future compared to ([-469, -221]), there is no way to find such a ( t ) that satisfies the rule. ### Conclusion Given the data ( A@[10,12] ) and the rule ( A leftarrow Diamondplus[4,13]A ), we cannot derive ( A@[-469,-221] ). Therefore, ( A@[-469,-221] ) is **not** true. |
| User Prompt | Based on your previous response, without any explanation state your answers to the question. You should only output "true" or "false" |
| LLM's output | false |
| Expected Answer | true |

False Positive Example:

| | |
|---|---|
| System Prompt | You are given a dataset and a temporal rule, and your task is to judge whether the given fact is entailed by the dataset and the rule. The rules are expressed as DatalogMTL, a knowledge representation language that extends Datalog with operators from metric temporal logic (MTL). The semantics of four MTL operators are given as follows:If Diamondminus[a,b]A is true at the time t, it requires that A needs to be true at some time between t-b and t-a.If Boxminus[a,b]A is true at the time t, it requires that A needs to be true continuously between t-b and t-a.If Diamondplus[a,b]A is true at the time t, it requires that A needs to be true at some point between t+a and t+b.If Boxplus[a,b]A is true at the time t, it requires that A needs to be true continuously between t+a and t+b. |
| User Prompt | Now we have some temporal data and some rules, data: A@[2,2] 
 rule: A:-Diamondminus[4,15]A 
 Is A@[-24,25] true or not? Do not answer directly, think step by step. |
| LLM's output | To determine whether ( A@[-24,25] ) is true given the data and the rule, we need to analyze the rule and the data step by step. 
 ### Step-by-Step Analysis 
 1. **Understanding the Rule:** - The rule is ( A leftarrow Diamondminus[4,15]A ). - This means that ( A ) is true at time ( t ) if ( A ) was true at some time between ( t-15 ) and ( t-4 ). 
 2. **Given Data:** - ( A@[2,2] ) means ( A ) is true at time 2. 
 3. **Determine the Time Intervals:** - We need to check if ( A ) can be true at any time ( t ) within the interval ([-24, 25]). 
 4. **Applying the Rule:** - For ( A ) to be true at any time ( t ) within ([-24, 25]), there must be some time ( t' ) within ([t-15, t-4]) where ( A ) is true. 
 5. **Checking the Data:** - We know ( A ) is true at time 2. - We need to find if there exists any ( t ) in ([-24, 25]) such that ( 2 ) lies within ([t-15, t-4]). 
 6. **Finding the Valid ( t ):** - For ( 2 ) to be within ([t-15, t-4]): [ t-15 leq 2 leq t-4 ] - Solving these inequalities: [ t-15 leq 2 implies t leq 17 ] [ 2 leq t-4 implies t geq 6 ] - Therefore, ( t ) must be in the interval ([6, 17]). 
 7. **Checking the Interval ([-24, 25]):** - The interval ([6, 17]) is within the interval ([-24, 25]). 
 ### Conclusion Since there exists a time ( t ) within ([-24, 25]) (specifically within ([6, 17])) where the rule can be satisfied by the given data, ( A@[-24,25] ) is true. |
| User Prompt | Based on your previous response, without any explanation state your answers to the question. You should only output "true" or "false" |
| LLM's output | true |
| Expected Answer | false |

## E   CAN LLM UNDERSTAND SYMBOLIC REPRESENTATION?

We ask LLM to translate generated natural language representations of logic rules to symbolic representations and them compare the translated symbolic representations with the ground truth symbolic representation to verify if LLM has the ability to understand symbolic representations. Specifically, we passed the same prompt that used in our symbolic evaluations, "The rules are expressed as DatalogMTL, a......If Boxplus[a,b]A is true at the time t, it requires that A needs to be true continuously between t+a and t+b.", into LLMs, along with few examples telling LLM the output format, then ask LLM to translate verbalized samples into symbolic ones. If the translated symbolic rule from the verbalized rule is exactly the same as the original symbolic rule, then we consider LLM has the ability to understand both the symbolic rule and the verbalized rule.

We passed 50 samples selected from **MultiRules** subset, which is considered the most challenging, into the LLMs. LLM accurately translated 96% of testing samples from verbalized representations to symbolic representations, so we consider that LLM can understand the symbolic representation.

# F DETAILED BENCHMARK CONSTRUCTION PSEUDO CODE

Our dataset generation algorithm is driven by generating rules. In a high level view, it generate rules one by one in a same context, while the generation process for each rule contains the context check, ensuring the generated rules are non-trivial.

---

**Algorithm 1:** Generate

---

**Parameters:** $f$: The set of features Enabled
**Parameters:** $N$: The number of rules
**Parameters:** $V$: A boolean flag to control if the program should generate a positive sample or a negative sample
**Output:** A problem instance $I$ containing a set of rules, a set of data, a query and a boolean value representing whether the query is valid or not.
$G \leftarrow EmptyGraph()$;
**while** $i$ in 1.....N **do**
   **do**
      $G \leftarrow GenerateGraph(G)$;
      **while** $n$ in $G.nodes$ **do**
         | Assign node with random values
      **end**
      $G \leftarrow GenerateRules(G)$
   **while** *New Info can be Inferred from I*;
**end**
$Rules, Data \leftarrow Extract\ Rules\ associated\ with\ G$;
$DeltaNew \leftarrow Facts\ Inferred\ From\ Graph\ G$;
$QueryEntity, Interval \leftarrow$ Randomly Select From $DeltaNew$;
**if** *V* **then**
   | $QueryInterval \leftarrow$ A random sub-interval from $Interval$;
**else**
   | $QueryInterval \leftarrow$ A random sub-interval that is not in $Interval$;
**end**
**return** *Rules, Data, QueryEntity, QueryInterval, V*

---

The graph generation algorithm 2 will generate a graph where nodes in the graph represents predicates such as A, B and C. We are going to attach details information about predicates and rules into the corresponding nodes and edges of the graph, but at this time we only need the structure of the graph, i.e. nodes and edges don't have special information attached.

**Algorithm 2:** Graph Generation

---

**Input:** $G$: The existing graph
**Parameters:** $f$: The set of features Enabled
**Output:** $G$: The generated graph (including the old information in the existing graph)
**Output:** $List[V]$: The list of new nodes, representing predicates, in the new graph
**Output:** $V_o$: The output node which depends on the some other nodes (in case that recursive is
        not enabled in $f$) in $List[V]$
$NewNode \leftarrow []$
Determine the lowest possible number of nodes to add $l$ and the highest number of possible
 nodes to add $r$ based on $f$.
$N \leftarrow random(l, r)$;
**while** $i$ in 1.....N **do**
    $p \leftarrow A\ randomly\ assigned\ predicate$;
    $G.AddNode(p)$;
    $NewNode.Push(p)$;
**end**
$OutNode \leftarrow RandomSelect(NewNode)$;
**while** $p$ in NewNode **do**
    **if** *"recursive" not in f and p == OutNode* **then**
        continue;
    **end**
    $G.AddEdge(p, OutNode)$
**end**
**return** *G, NewNode, OutNode*

---

After the structure of the graph is generated, we are going to attach rule information to each edge of the graph using the Rule Generation algorithm 3. Since we are doing Graph Generation and Rule Generation alternately, in the rule generation we only care about edges that don't already has a rule, we will skip the edges that already has a rule associated with that.

**Algorithm 3:** Rule Generation

---

**Input:** $G$: The existing graph
**Parameters:** $f$: The set of features Enabled
**Output:** $G$: The generated graph (including the old information in the existing graph)
$SelectedOp \leftarrow Set()$
$SelectedOp.add(RandomSelect(Boxminus, Boxplus, Diamondplusm, Diamondminus))$
**if** *"mixed_operators" in f* **then**
    Randomly select and add more operators to SelectedOp;
**end**
**while** $Edge$ in G **do**
    $u, v, a \leftarrow G$;
    **if** *No rule is associated with Edge* **then**
        $Op \leftarrow$ Randomly select an operator from $SelectedOp$;
        $Interval \leftarrow$ Randomly create an interval;
        Create an item literal with $Op$ and $Interval$ and associated that with $Edge$;
    **end**
**end**
**return** $G$

---

