# OpenReview forum: "tBen: Benchmarking and Testing the Rule-Based Temporal Logic Reasoning Ability of Large Language Models with DatalogMTL"
_ICLR.cc/2025/Conference — Submitted to ICLR 2025_

### Official Review · Reviewer_dT7E · 2024-10-16

**Soundness:** 3
**Presentation:** 3
**Contribution:** 2
**Rating:** 3
**Confidence:** 4

**Summary:**

This paper proposes TBen, a benchmark for evaluating the ability of LLMs to perform rule-based temporal logic reasoning.
TBen uses DatalogMTL to describe problems, a language capable of describing temporal logic data.
TBen defines six levels of rule complexity, allowing customization of reasoning data formats based on logical complexity and temporal rules.
The authors conduct experiments with standard prompting techniques and chain-of-thought prompting techniques on GPT-4o and LLaMA3 models to explore the models' performance on temporal logic reasoning.
The results show that the models only achieve results similar to random guessing when generating intermediate steps.
Additionally, this paper finds that current models are unable to solve temporal reasoning problems involving recursive forms, indicating the challenges faced by current models in solving complex temporal logic reasoning.

**Strengths:**

1. This paper proposes tBen for evaluating the ability of large language models to perform rule-based temporal logic reasoning. Furthermore, the data construction approach in the paper allows for the adjustment of the types of time and the complexity of logic to generate benchmarks at different difficulty levels.
2. Evaluations are conducted based on the proposed benchmark with state-of-the-art LLMs. They analyze the shortcoming and challenges faced by current LLMs.
3. The authors also conduct some analysis on some topics, such as rule v.s. NL and ablations.

**Weaknesses:**

1. The TBen benchmark is entirely rule-generated and can yield deterministic results using solvers, which limits the breadth of its application. This benchmark is only suitable for evaluating models' performance in rule-based temporal logic reasoning and cannot be extended to certain real-world scenarios. The dataset's usage scenarios are significantly restricted, which is a shortcoming in its content. In other words, this benchmark is more tailored to serve the analysis testbed for this paper rather than subsequent follow-up research.
2. If this paper is regarded as an evaluation paper, then the analysis work and related in-depth discussions conducted in this paper are insufficient. The paper only conducted experiments on two models and the analysis was relatively superficial, lacking in-depth analysis.

**Questions:**

1. This benchmark is fully symbolic, so whether there will be deviations in the process of converting it into natural language form (verbalization), leading to a gap between language models' handling of symbolic reasoning and natural language reasoning.

---

> ### Author Response · Authors · 2024-11-24
>
> > The TBen benchmark is entirely rule-generated and can yield deterministic results using solvers, which limits the breadth of its application. This benchmark is only suitable for evaluating models' performance in rule-based temporal logic reasoning and cannot be extended to certain real-world scenarios. The dataset's usage scenarios are significantly restricted, which is a shortcoming in its content. In other words, this benchmark is more tailored to serve the analysis testbed for this paper rather than subsequent follow-up research.
>
> Thank you for raising the concern. The LLM's temporal reasoning ability is transferable. If it exhibts good termporal reasoning in logic form, it's likely to be able to transfer the ability to many down stream tasks. For instance, [1] highlights that training the LLM component of a video understanding foundation model on temporally focused textual data can enhance its overall video reasoning performance. Thus, improving LLMs’ temporal reasoning abilities through our benchmark has the potential to significantly benefit downstream applications requiring this skill.
>
> The other advantage of primarily using symbolc representation is that through a controllable temporal data generator, we can construct the dataset automatically, ensuring that the dataset is scalable and significantly less prone to issues such as data leakage, a common concern in benchmark creation.
>
>
> [1] Li, L., Liu, Y., Yao, L., Zhang, P., An, C., Wang, L., ... & Liu, Q. (2024). Temporal Reasoning Transfer from Text to Video. arXiv preprint arXiv:2410.06166.
>
>
> > If this paper is regarded as an evaluation paper, then the analysis work and related in-depth discussions conducted in this paper are insufficient. The paper only conducted experiments on two models and the analysis was relatively superficial, lacking in-depth analysis.
>
> Thank you for your valuable feedback. Our primary objective is to evaluate the temporal reasoning capabilities of various LLMs, encompassing both open-source and proprietary models. While the initial version of our paper reports results from one representative model of each type, we recognize the importance of expanding the analysis to provide a more comprehensive evaluation. In the final version, we will include results from additional models to strengthen the breadth and depth of our analysis. This will also allow us to offer a more detailed discussion and nuanced insights into the temporal reasoning performance across diverse LLMs.
>
> > This benchmark is fully symbolic, so whether there will be deviations in the process of converting it into natural language form (verbalization), leading to a gap between language models' handling of symbolic reasoning and natural language reasoning.
>
> Thank you for raising this insightful concern. We have addressed this issue in multiple ways within our work. First, we manually inspected several CoT contexts and did not observe any evidence suggesting that LLMs misunderstand the symbolic format. Second, as presented in Appendix E, we conducted experiments to evaluate whether LLMs can reliably convert between natural language and symbolic formats. These experiments demonstrated that the models are capable of handling the symbolic format accurately. Third, we tested the models directly on natural language representations and observed similar performance levels to those with the symbolic format. Together, these findings provide strong evidence that the models can comprehend the symbolic format under the introductory prompt we provided, indicating that the observed limitations are not due to format unfamiliarity but likely stem from the inherent challenges of temporal reasoning.

---

> > ### Comment · Reviewer_dT7E · 2024-11-26
> > **Response to authors**
> >
> > Thank you for the feedback; I have read the author's response.
> > However, I believe that the substantive issues in the article, such as lack of analysis, the overly symbolic nature of the benchmark, still require future revisions to address. Therefore, I have decided to maintain my rating.

---

### Official Review · Reviewer_DBvf · 2024-11-03

**Soundness:** 3
**Presentation:** 3
**Contribution:** 1
**Rating:** 3
**Confidence:** 4

**Summary:**

The paper introduces TBEN, a new synthetic benchmark designed to evaluate the rule-based temporal logic reasoning abilities of large language models (LLMs). TBEN focuses on DatalogMTL, an automated reasoning language that extends Datalog with temporal logic, enabling reasoning about time. The benchmark consists of tasks spanning six levels of complexity, from simple single-atom rules to more complex recursive rules.

The authors evaluate two LLMs (GPT-4 and Llama-3) using three prompting strategies: zero-shot, few-shot, and zero-shot chain-of-thought (CoT). They find that without reasoning (i.e., CoT), even advanced models perform poorly. They also observe that both models struggle with recursive reasoning, highlighting limitations in handling advanced temporal logic problems.

**Strengths:**

- **Importance of temporal reasoning** Reasoning about time is a key aspect of intelligence, which seems to be still understudied in the area of large language models. The paper shows (to an extent) how LLMs struggle in this task.

- **Well thought task generation** The authors present a well-thought method for generating synthetic tasks with controlled/varying levels of complexity. Such systematic approach is valuable.

**Weaknesses:**

- **Logical Reasoning or Language Familiarity?**: It is unclear whether the poor performance of LLMs is due to their inability to reason about temporal concepts or their unfamiliarity with the formal syntax and semantics of DatalogMTL. The benchmark may conflate temporal reasoning abilities with decoding a niche formal language: the evaluated models may not have been adequately exposed to formal temporal logic during training (actually even in natural language, LLMs tend to perform worse on languages with lesser training data). Not referenced in the text, Appendix E touches a bit this aspect, but is not sufficient (besides being not well written yet).


- **Impact on the Broader Community**: The practical implications and impact of the findings on the ICLR community and AI research at large are limited (beyond showing another domain where LLMs struggle). This for me the most crucial weakness. To be suitable for ICLR, the paper should have may be tested the temporal reasoning in truly natural language contexts (not just templated translations of a formal language like Figure 4.) and provided deep insights on why they work or fail at these tasks etc. In its current form, the paper is a better fit for a specialized benchmark workshop.

- **Binary question format**: The tasks result in simple true/false answers. So while this eases the task of the evaluation, such binary questions may not capture the full complexity of temporal reasoning and could limit the depth of insights gained from the evaluation.

- **Clarity and Referencing Issues**: The manuscript frequently references the "Appendix" without specifying sections or page numbers, making it challenging for readers to locate supplementary information.

**Questions:**

In which way exposing the formal temporal reasoning weaknessess in LLMs is essential for the broader goal of improving reasoning abilities of LLMs? How does it differ from exposing weaknesses in any other specific domain like coding in a particular  programming language?

How do we know if we're testing the models abilities to reason about time?  or if we're limited by the models' lack of fluency in this niche formal language to which the models were not as much exposed as the rest?

Why not put all the focus on the natural language temporal reasoning abilities of the models? since natural language is what they were exposed to most?

As a follow-up, the current templated way of generating natural language questions is good, but "unnatural", how about using LLMs to generate the natural language text instead? and transforming the entire exercise into an actual natural language based reasoning about time?

Finally, it would be interesting to extend the evaluation to the "large reasoning models" like the open AI's o1 family.

---

> ### Author Response · Authors · 2024-11-24
>
> > Logical Reasoning or Language Familiarity
>
> Thank you for raising this insightful concern. We have addressed this issue in multiple ways within our work. First, we manually inspected several CoT contexts and did not observe any evidence suggesting that LLMs misunderstand the symbolic format. Second, as presented in Appendix E, we conducted experiments to evaluate whether LLMs can reliably convert between natural language and symbolic formats. These experiments demonstrated that the models are capable of handling the symbolic format accurately. Third, we tested the models directly on natural language representations and observed similar performance levels to those with the symbolic format. Together, these findings provide strong evidence that the models can comprehend the symbolic format under the introductory prompt we provided, indicating that the observed limitations are not due to format unfamiliarity but likely stem from the inherent challenges of temporal reasoning.
>
> > Impact on the Broader Community
>
> We are particularly interested in leveraging LLMs for temporal reasoning tasks because the capabilities LLMs develop in these areas can be transferred to various practical downstream applications. Temporal reasoning, in particular, represents a critical bottleneck for many tasks, especially in the multimodal domain, such as video understanding. For instance, [1] highlights that training the LLM component of a video understanding foundation model on temporally focused textual data can enhance its overall video reasoning performance. Thus, improving LLMs’ temporal reasoning abilities through our benchmark has the potential to significantly benefit downstream applications requiring this skill.
>
>
> [1] Li, L., Liu, Y., Yao, L., Zhang, P., An, C., Wang, L., ... & Liu, Q. (2024). Temporal Reasoning Transfer from Text to Video. arXiv preprint arXiv:2410.06166.
>
> > Binary question format
>
> Thank you for raising this insightful concern. We would like to address it by highlighting that many complex problems, such as optimization problems, can be effectively transformed into equivalent decision problems. Decision problems, including Yes/No questions, provide a foundational means to evaluate an LLM's capability in temporal logical reasoning. Achieving strong performance on decision problems is therefore indicative of the model's ability to handle the underlying reasoning tasks, including metric temporal reasoning. While descriptive questions can offer additional insights, decision problems serve as a rigorous and minimalistic approach to benchmark reasoning skills systematically and comprehensively.
>
> > Clarity and Referencing Issues
>
> Thank you for the suggestion. We will include those refernces into our main text in our final versin.

---

> > ### Comment · Reviewer_DBvf · 2024-11-27
> >
> > Thank you for your response. While I appreciate the efforts, I don't think they address my main concern. The most important question isn't whether LLMs can handle the formal notation - it's whether focusing on a niche formal language (DatalogMTL) is the right approach to understanding temporal reasoning capabilities. Natural language temporal reasoning is a more urgent and fundamental challenge that would benefit the broader ICLR community. The video understanding example is interesting, but it actually reinforces my point about prioritizing natural language temporal understanding over formal systems. Thank you again for engaging with my concerns.

---

> > > ### Author Response · Authors · 2024-11-28
> > >
> > > Dear Reviewer DBvf,
> > >
> > > Thanks for your reading our response.
> > >
> > > We acknowledge the importance of natural language temporal reasoning and recognize the extensive efforts being made in this area. However, this paper approaches the topic from a different perspective, focusing on the temporal reasoning capabilities of LLMs in symbolic contexts—a field that has been studied extensively in traditional AI domains. While I respectfully disagree with the notion that this work is less valuable due to a suggested prioritization of natural language temporal understanding over formal systems, I believe that examining temporal reasoning in symbolic contexts remains equally significant and meaningful.
> > >
> > > Best regards,
> > > Authors

---

### Official Review · Reviewer_fUzn · 2024-11-04

**Soundness:** 2
**Presentation:** 3
**Contribution:** 2
**Rating:** 5
**Confidence:** 3

**Summary:**

The paper constructs a benchmark for evaluating the rule-based temporal reasoning capabilities of large language models, using a knowledge representation language called DatalogMTL. It introduces the DatalogMTL language, outlines the dataset construction process, and provides details on the final evaluation tasks. Besides, It tests models such as GPT-4o and Llama-3 on this dataset.

**Strengths:**

1. Unlike previous works that focus more on natural language-based logical reasoning abilities, this paper evaluates the rule-based temporal logic reasoning abilities of large language models.
2. This paper provides a detailed introduction to the representation language DatalogMTL and the entire benchmark construction process, which will help readers gain a better understanding of the evaluation benchmark.

**Weaknesses:**

1. This paper should provide a more detailed discussion on the significance of this dataset. For the LLMs community, what are the core challenges that this dataset presents compared to other benchmarks assessing the logic reasoning abilities of large language models? Since it ultimately translates into prompts for the models, what is the unique significance of temporality in this context? Does it introduce challenges that are completely different from previous tasks in logic, mathematics, and code reasoning? On another note, how can performing well on this benchmark assist in downstream tasks? Additionally, for the traditional rule-based reasoning community, why should LLMs be applied to this rule-based temporal task, and what are the advantages and disadvantages of LLMs compared to traditional rule-based reasoning methods?
2. The current design of the prompts seems inadequate. For example, in line 746, it does not explain to the model what the "@" symbol means, so it is unreasonable to expect the model to produce correct results. Additionally, in Figure 2, how does the model understand the meanings of these combinations of expressions? The existing prompts do not seem to clearly explain this to the model.

**Questions:**

Please refer to the weaknesses section above.

**Details Of Ethics Concerns:**

No concerns.

---

> ### Author Response · Authors · 2024-11-24
>
> > This paper should provide a more detailed discussion on the significance of this dataset. For the LLMs community, what are the core challenges that this dataset presents compared to other benchmarks assessing the logic reasoning abilities of large language models? Since it ultimately translates into prompts for the models, what is the unique significance of temporality in this context? Does it introduce challenges that are completely different from previous tasks in logic, mathematics, and code reasoning? On another note, how can performing well on this benchmark assist in downstream tasks?  Additionally, for the traditional rule-based reasoning community, why should LLMs be applied to this rule-based temporal task, and what are the advantages and disadvantages of LLMs compared to traditional rule-based reasoning methods?
>
> Compared to existing work, the primary challenge we address lies in constructing a controllable temporal data generator. This approach ensures that the dataset is scalable and significantly less prone to issues such as data leakage, a common concern in benchmark creation. By achieving this, we provide a robust platform for evaluating LLMs on temporal reasoning tasks.
>
> We are particularly interested in leveraging LLMs for rule-based reasoning tasks rather than relying on traditional methods because the capabilities LLMs develop in these areas can be transferred to various practical downstream applications. Temporal reasoning, in particular, represents a critical bottleneck for many tasks, especially in the multimodal domain, such as video understanding. For instance, [1] highlights that training the LLM component of a video understanding foundation model on temporally focused textual data can enhance its overall video reasoning performance. Thus, improving LLMs’ temporal reasoning abilities through our benchmark has the potential to significantly benefit downstream applications requiring this skill.
>
>
> [1] Li, L., Liu, Y., Yao, L., Zhang, P., An, C., Wang, L., ... & Liu, Q. (2024). Temporal Reasoning Transfer from Text to Video. arXiv preprint arXiv:2410.06166.
>
> > The current design of the prompts seems inadequate. For example, in line 746, it does not explain to the model what the "@" symbol means, so it is unreasonable to expect the model to produce correct results. Additionally, in Figure 2, how does the model understand the meanings of these combinations of expressions? The existing prompts do not seem to clearly explain this to the model.
>
> We appreciate the reviewer’s thorough inspection of the provided examples. We acknolwedge that that "@" symbol is not directly explanined in the prompt, but  this is not likely to have impact on our final results since the meaning of "@" is straight forward and there is evidence that LLM is able to understand it. In L827, we can see that LLM responded as "Now, let’s look at the data: B@[3,10]. This means that B is true at some point between time 3 and time 10.", proving that it understand B is true at some point between time 3 and time 10.
>
> Generally, there are lots of evidence proving that LLM understands the symbolic representation very well. First, we manually inspected several CoT contexts and did not observe any evidence suggesting that LLMs misunderstand the symbolic format. Second, as presented in Appendix E, we conducted experiments to evaluate whether LLMs can reliably convert between natural language and symbolic formats. These experiments demonstrated that the models are capable of handling the symbolic format accurately. Third, we tested the models directly on natural language representations and observed similar performance levels to those with the symbolic format. Together, these findings provide strong evidence that the models can comprehend the symbolic format under the introductory prompt we provided, indicating that the observed limitations are not due to format unfamiliarity but likely stem from the inherent challenges of temporal reasoning.

---

> > ### Comment · Reviewer_fUzn · 2024-11-26
> > **Reply to the authors**
> >
> > Thank you for the response first!
> >
> > Could you please provide a more detailed response to the following two mentioned concerns?
> >
> > > (Uniqueness) Since it ultimately translates into prompts for the models, what is the unique significance of temporality in this context? Does it introduce challenges that are completely different from previous tasks in logic, mathematics, and code reasoning?
> >
> > > (Generalization) On another note, how can performing well on this benchmark assist in downstream tasks? (we may need more specific reasoning regarding this benchmark, such as how making progress on this benchmark can be transferred to and assist a broader range of downstream tasks, rather than simply stating that improving the model's overall temporal capabilities will enable it to better handle other tasks)
> >
> > Best,
> > Reviewer fUzn

---

> > > ### Author Response · Authors · 2024-11-27
> > >
> > > Thank you for your suggestion.
> > >
> > > > Uniqueness
> > >
> > > Compared to other reasoning benchmarks such as Boolean logic, mathematics, or code reasoning/generation, the unique significance of temporal reasoning lies in its dual requirements: logical reasoning and basic numerical reasoning. Unlike logic or code generation benchmarks, temporal reasoning demands the ability to perform straightforward numerical operations such as comparing two numbers, addition, and subtraction. In contrast to mathematical reasoning benchmarks, which often focus on intricate and abstract mathematical problems, tBen emphasizes intuitive mathematics, limited to simple comparison, addition, and subtraction.
> > >
> > > This distinction is intentional and reflects the nature of human reasoning in daily life, where we frequently combine logical reasoning with basic numerical calculations, but rarely engage in complex mathematical reasoning. Thus, our approach aligns with the types of reasoning commonly used in real-world scenarios.
> > >
> > > > Generalization
> > >
> > > According to [1] , temporal data plays a critical role across a wide range of applications, including stock trading [2], network flow anomaly detection [3], and equipment malfunction monitoring [4]. DatalogMTL, as a robust Knowledge Representation (KR) language, demonstrates exceptional generalization ability by effectively addressing diverse temporal reasoning tasks. Its versatility has been highlighted in various domains, such as ontology-based query answering [5] and stream reasoning [6], showcasing its capacity to unify and extend reasoning methodologies across temporal data challenges.
> > >
> > > [1] Wang, D., Hu, P., Wałęga, P. A., & Grau, B. C. (2022, June). Meteor: Practical reasoning in datalog with metric temporal operators. In Proceedings of the AAAI Conference on Artificial Intelligence (Vol. 36, No. 5, pp. 5906-5913).
> > >
> > > [2] Nuti, G.; Mirghaemi, M.; Treleaven, P.; and Yingsaeree, C. 2011. Algorithmic trading. Computer, 44(11): 61–69.
> > >
> > > [3] Munz, G.; and Carle, G. 2007. Real-time analysis of flow data for
> > >
> > > [4] Doherty, P.; Kvarnström, J.; and Heintz, F. 2009. A temporal logic-based planning and execution monitoring framework for unmanned aircraft systems. Proc. of AAMAS, 19(3): 332–377.
> > >
> > > [5] Brandt, S.; Kalaycı, E. G.; Ryzhikov, V.; Xiao, G.; and Zakharyaschev, M. 2018. Querying log data with metric temporal logic. J. Artif. Intell. Res., 62: 829–877.
> > >
> > > [6] Wałega, P. A.; Cuenca Grau, B.; Kaminski, M.; and Kostylev, E. V. 2019. DatalogMTL: computational complexity and expressive power. In Proc. of ĲCAI, 1886–1892.

---

> > > > ### Comment · Reviewer_fUzn · 2024-11-29
> > > > **Reply to the authors**
> > > >
> > > > Thank you for your further explanations!
> > > >
> > > > While I still have reservations about the overall importance of this benchmark to the community (for example, whether it truly warrants significant attention given the variety of benchmarks currently available), your responses have addressed my concerns to some extent. Therefore, I have updated my score.
> > > >
> > > > Best,
> > > > Reviewer fUzn

---

### Official Review · Reviewer_vhgk · 2024-11-08

**Soundness:** 2
**Presentation:** 3
**Contribution:** 2
**Rating:** 5
**Confidence:** 4

**Summary:**

The paper presents a synthetic benchmark to evaluate temporal logic reasoning capabilities of large language models (LLMs). The benchmark uses the DatalogMTL language, to represent problems involving rule-based temporal logic. The evaluation on GPT-4o and LLama3 show that LLMs perform poorly on this benchmark highlighting a gap in LLM capabilities when compared to traditional symbolic approaches.

**Strengths:**

The authors look at metric temporal reasoning problems (specifically fact entailment ones) and provide a varied set of problems to evaluate LLMs. They evaluate both an open-source and a closed model and provide ablation studies to get better understanding of the LLMs’ capabilities in such temporal reasoning tasks.

**Weaknesses:**

My primary concern with the proposed benchmark is that it relies on Yes or No questions, which I believe are inadequate for effectively measuring the logical reasoning abilities of an LLM. For a more informative evaluation of whether LLMs are capable of metric temporal reasoning, the benchmark should employ more descriptive questions.

There have been several works that examine LLM performance on traditional reasoning problems, such as satisfiability, PDDL planning, and scheduling [1,2,3], and these studies consistently show that LLMs perform poorly on such tasks. Additionally, other works have highlighted the inability of Chain of Thought to generalize within such reasoning problems [4,5]. However, these works have not been compared or contrasted to in the paper in order to better position the benchmark in terms of its contribution.

[1] Ye, Xi, Qiaochu Chen, Isil Dillig, and Greg Durrett. "Satlm: Satisfiability-aided language models using declarative prompting." Advances in Neural Information Processing Systems 36 (2024).

[2] Valmeekam, Karthik, Matthew Marquez, Alberto Olmo, Sarath Sreedharan, and Subbarao Kambhampati. "Planbench: An extensible benchmark for evaluating large language models on planning and reasoning about change." Advances in Neural Information Processing Systems 36 (2024).

[3] Silver, Tom, Varun Hariprasad, Reece S. Shuttleworth, Nishanth Kumar, Tomás Lozano-Pérez, and Leslie Pack Kaelbling. "PDDL planning with pretrained large language models." In NeurIPS 2022 foundation models for decision making workshop. 2022.

[4] Dziri, Nouha, Ximing Lu, Melanie Sclar, Xiang Lorraine Li, Liwei Jiang, Bill Yuchen Lin, Sean Welleck et al. "Faith and fate: Limits of transformers on compositionality." Advances in Neural Information Processing Systems 36 (2024).

[5] Stechly, Kaya, Karthik Valmeekam, and Subbarao Kambhampati. "Chain of thoughtlessness: An analysis of cot in planning." arXiv preprint arXiv:2405.04776 (2024).

**Questions:**

Are there details on the accuracy of the COTs generated by the LLMs?

---

> ### Author Response · Authors · 2024-11-24
>
> > My primary concern with the proposed benchmark is that it relies on Yes or No questions, which I believe are inadequate for effectively measuring the logical reasoning abilities of an LLM. For a more informative evaluation of whether LLMs are capable of metric temporal reasoning, the benchmark should employ more descriptive questions.
>
> Thank you for raising this insightful concern. We would like to address it by highlighting that many complex problems, such as optimization problems, can be effectively transformed into equivalent decision problems. Decision problems, including Yes/No questions, provide a foundational means to evaluate an LLM's capability in temporal logical reasoning. Achieving strong performance on decision problems is therefore indicative of the model's ability to handle the underlying reasoning tasks, including metric temporal reasoning. While descriptive questions can offer additional insights, decision problems serve as a rigorous and minimalistic approach to benchmark reasoning skills systematically and comprehensively.
>
> > There have been several works that examine LLM performance on traditional reasoning problems, such as satisfiability, PDDL planning, and scheduling [1,2,3], and these studies consistently show that LLMs perform poorly on such tasks. Additionally, other works have highlighted the inability of Chain of Thought to generalize within such reasoning problems [4,5]. However, these works have not been compared or contrasted to in the paper in order to better position the benchmark in terms of its contribution.
>
> Thank you for bringing these works to our attention. While we appreciate their contributions, our work is fundamentally different from the studies mentioned. For example:
>
> * [1] focuses on static logical reasoning without incorporating temporal information, whereas our work centers on temporal reasoning.
> * [2] and [3] adopt discretized time steps but do not evaluate numerical reasoning, which is a critical aspect of our temporal reasoning benchmark.
> * [4] considers only three specific types of temporal reasoning tasks, while our work introduces a more general syntax capable of representing these and a broader range of problems.
> * [5] is limited to a particular type of temporal reasoning problem, reducing its overall diversity.
>
> Most notably, contrary to the conclusions drawn in prior works, we found that Chain of Thought (CoT) prompting can significantly enhance LLMs' temporal reasoning capabilities. This provides a novel insight into the role of CoT in reasoning tasks and highlights the distinct contributions of our benchmark.
>
> We will include these works in the related work section of the final version of our paper to better contextualize and position our contributions.
> > Are there details on the accuracy of the COTs generated by the LLMs?
>
> We have included those information in Figure 5.

---

### Meta-Review · Area_Chair_biiD · 2024-12-21

**Metareview:**

The paper presents a synthetic benchmark to evaluate temporal logic reasoning capabilities of LLMs. Whether LLMs indeed possess the ability to reason and to what to extent is an important question in the community and is gaining more and more attention. Reviewers all appreciate the importance of this direction. However, many benchmarks are proposed in the recent years. All the reviewers in some degree raise the concern about what is the unique value of this benchmark compared to other rule-based logic benchmarks. Besides testing some LLMs, reviewers hope, and I agree, that we would like to gain some further insights from those experiments. What weakness of LLMs does this benchmark expose, and how addressing this weakness can improve LLM's reasoning in general. More discussion on more experiments that provide insights of why LLMs fail on reasoning would definitely make this paper stronger. Hence, I recommend a rejection.

**Additional Comments On Reviewer Discussion:**

The clarity issues are addressed by the rebuttal (e.g., what certain symbol means in the benchmark). However, almost all reviewers raised the concern about the importance of this benchmark, and how this benchmark is different from other reasoning benchmarks, more importantly, the analysis of the results. And those concerns still remain after the rebuttal.

---

### Decision · Program_Chairs · 2025-01-22

Reject